# Characterization of "dead-zone" eddies in the eastern tropical North Atlantic

**Florian Schütte [1], Johannes Karstensen [1], Gerd Krahmann [1], Helena Hauss [1], Björn Fiedler [1],**

**Peter Brandt [1,2], Martin Visbeck [1,2] and Arne Körtzinger [1,2]**

[1] GEOMAR Helmholtz Centre for Ocean Research Kiel, Kiel, Germany

[2] Christian-Albrechts-Universität zu Kiel, Kiel, Germany

*Correspondence to*: F. Schütte (fschuette@geomar.de)

**Abstract**

Localized open-ocean low–oxygen "dead-zones" in the eastern tropical North Atlantic are recently discovered ocean features that can develop in dynamically isolated water masses within cyclonic eddies (CE) and anticyclonic modewater eddies (ACME). Analysis of a comprehensive oxygen dataset obtained from gliders, moorings, research vessels and Argo floats reveals that "dead-zone" eddies are found in surprisingly high numbers and in a large area from about 4°N to 22°N, from the shelf at the eastern boundary to 38°W. In total, 173 profiles with oxygen concentrations below the minimum background concentration of 40 µmol kg$^{-1}$ could be associated with 27 independent eddies (10 CEs; 17 ACMEs) over a period of 10 years. Lowest oxygen concentrations in CEs are less than 10 µmol kg$^{-1}$ while in ACMEs even suboxic (< 1 µmol kg$^{-1}$) levels are observed. The oxygen minimum in the eddies is located at shallow depth from 50 to 150 m with a mean depth of 80 m. Compared to the surrounding waters, the mean oxygen anomaly in the core depth range (50 and 150 m) for CEs (ACMEs) is -38 (-79) µmol kg$^{-1}$. North of 12°N, the oxygen depleted eddies carry anomalously low salinity water of South Atlantic origin from the eastern boundary upwelling region into the open ocean. Here water mass properties and satellite eddy tracking both point to an eddy generation near the eastern boundary. In contrast, the oxygen depleted eddies south of 12°N carry weak hydrographic anomalies in their cores and seem to be generated in the open ocean away from the boundary. In both regions a decrease in oxygen from east to west is identified supporting the en-route creation of the low-oxygen core through a combination of high productivity in the eddy surface waters and an isolation of the eddy cores with respect to lateral oxygen supply. Indeed, eddies of both types feature a cold sea surface temperature anomaly and enhanced chlorophyll concentrations in their center. The low-oxygen core depth in the eddies aligns with the depth of the shallow oxygen minimum zone of the eastern tropical North Atlantic. Averaged over the whole area an oxygen reduction of 7 µmol kg$^{-1}$ in the depth range of 50 to 150 m (peak reduction is 16 µmol kg$^{-1}$ at 100 m depth) can be associated to the dispersion of the eddies. Thus the locally increased oxygen consumption within the eddy cores enhances the total oxygen consumption in the open eastern tropical North Atlantic Ocean and seem to be an contributor to the formation of the shallow oxygen minimum zone.

## 1. Introduction

The eastern tropical North Atlantic (ETNA: 4°N to 22°N and from the shelf at the eastern boundary to 38°W, Fig. 1) off Northwest Africa is one of the biologically most productive areas of the global ocean (Chavez and Messié, 2009; Lachkar and Gruber, 2012). In particular, the eastern boundary current system close to the Northwest African coast is a region where northeasterly trade winds force coastal upwelling of cold, nutrient rich waters, resulting in high productivity (Bakun, 1990; Lachkar and Gruber, 2012; Messié et al., 2009; Pauly and Christensen, 1995). The ETNA is characterized by a weak large-scale circulation and instead dominated by mesoscale variability (here referred to as eddies) (Brandt et al., 2015; Mittelstaedt, 1991). Traditionally the ETNA is considered to be "hypoxic", with minimal oxygen concentrations of marginally below 40 μmol kg$^{-1}$ (e.g. Stramma et al. (2009)) (Fig. 1a). The large-scale ventilation and oxygen consumption processes of thermocline waters in the ETNA result in two separate oxygen minima (Fig. 1b): a shallow one with a core depth of about 80 m and a deep one at a core depth of about 450 m (Brandt et al., 2015; Karstensen et al., 2008). The deep minimum is the core of the OMZ and is primarily created by sluggish ventilation of the respective isopycnals (Luyten et al., 1983; Wyrtki, 1962). It extends from the eastern boundary into the open ocean and is located in the so-called shadow zone of the ventilated thermocline, with the more energetic circulation of the subtropical gyre in the north and the equatorial region in the south (Karstensen et al., 2008; Luyten et al., 1983). The shallow oxygen minimum intensifies from the equator towards the north with minimal values near the coast at about 20°N (Brandt et al., 2015) (Fig. 1a). It is assumed that the shallow OMZ originates from enhanced biological productivity and an increased respiration associated with sinking particles in the water column (Brandt et al., 2015; Karstensen et al., 2008; Wyrtki, 1962).

The eddies act as a major transport agent between coastal waters and the open ocean (Schütte et al., 2016), which is a well-known process for all upwelling areas in the world oceans (Capet et al., 2008; Chaigneau et al., 2009; Correa-Ramirez et al., 2007; Marchesiello et al., 2003; Nagai et al., 2015; Schütte et al., 2016; Thomsen et al., 2015). In the ETNA, most eddies are generated near the eastern boundary, Rossby wave dynamics and the basin scale circulation force these eddies to propagate westwards (Schütte et al., 2016). Open ocean eddies with particularly high South Atlantic Central Water (SACW) fractions in their cores have been found far offshore in regions dominated by the much saltier North Atlantic Central Water (NACW) (Karstensen et al., 2015; Pastor et al., 2008). Weak lateral exchange across the eddy boundaries is most likely the reason for the isolation (Schütte et al., 2016). The impact of eddy transport on the coastal productivity (equivalent to other upwelling related properties) was investigated by Gruber et al. (2011), who were able to show that high (low) eddy driven transports of nutrient-rich water from the shelf into the open ocean results in lower (higher) biological production on the shelf. Besides acting as export agents for coastal waters and conservative tracers, coherent eddies have been reported to establish and maintain an isolated ecosystem changing non-conservative tracers with time (Altabet et al., 2012; Fiedler et al., 2016; Hauss et al., 2016; Karstensen et al., 2015; Löscher et al., 2015). Coherent/isolated mesoscale eddies can exist over periods of several months or even years (Chelton et al., 2011). During that time the biogeochemical conditions within these eddies can evolve very different from the surrounding water masses (Fiedler et al., 2016). Hypoxic to suboxic oxygen levels have been observed in cyclonic eddies (CEs) and anticyclonic modewater eddies (ACMEs) at shallow depth and just beneath the mixed layer (about 50 to 100 m) (Karstensen et al., 2015). The creation of the low-oxygen cores in the eddies have been attributed to the combination of several factors (Karstensen et al., 2015): high productivity in the surface waters of the eddy (Hauss et al., 2016; Löscher et al., 2015), enhanced respiration of sinking organic material at

subsurface depth (Fiedler et al., 2016; Fischer et al., 2016) and an "isolation" of the eddy core from exchange with surrounding and better oxygenated water (Karstensen et al., 2016). The intermittent nature of the oxygen depletion and the combination of high respiration with sluggish oxygen transport resamples what is known as "dead-zone" in other aquatic system (lakes, shallow bays), and therefore the term "dead-zone eddies" has been introduced (Karstensen et al., 2015). So far the profound impacts on behaviour of microbial (Löscher et al., 2015) and metazoan (Hauss et al., 2016) communities has been documented inside the eddies. For example, the appearance of denitrifying bacteria, typically absent from the open tropical Atlantic, has been observed (Löscher et al., 2015) via the detection of nirS gene transcripts (the key functional marker for denitrification). However, the close-to-Redfield N:P stoichiometry in ACMEs in the ETNA (Fiedler et al., 2016), does not suggest a large-scale net loss of bioavailable nitrogen via denitrification. The key point in changing non-conservative tracers in the eddy cores is the physical-biological coupling, which is strongly linked to the vertical velocities of submesoscale physics, stimulating primary production (upward nutrient flux) in particular under oligotrophic conditions (Falkowski et al., 1991; Levy et al., 2001; McGillicuddy et al., 2007). The detailed understanding of the physical and biogeochemical processes and their linkage in eddies is still limited (Lévy et al., 2012). Consequently the relative magnitude of eddy-dependent vertical nutrient flux, primary productivity and associated enhanced oxygen consumption or nitrogen fixation/denitrification in the eddy cores and continuously the contribution to the large-scale oxygen or nutrient distribution is fairly unknown.

In order to further investigate the physical, biogeochemical and ecological structure of "dead-zone" eddies, an interdisciplinary field study was carried out in winter 2013/spring 2014 in the ETNA, north of Cape Verde, using dedicated ship, mooring and glider surveys supported by satellite and Argo float data. The analysis of the field study data revealed surprising results regarding eddy meta-genomics (Löscher et al., 2015), zooplankton communities (Hauss et al., 2016), carbon chemistry (Fiedler et al., 2016) and nitrogen cycling (Karstensen et al., 2016). Furthermore, analyses of particle flux time series, using sediment trap data from the Cape Verde Ocean Observatory (CVOO), were able to confirm the impact of highly productive "dead-zone" eddies on deep local export fluxes (Fischer et al., 2016). In this paper we investigate "dead-zone" eddies detected from sea level anomaly (SLA) and sea surface temperature (SST) data based on methods described by Schütte et al. (2016). We draw a connection between the enhanced consumption and associated low-oxygen concentration in eddy cores and the formation of the regional observed shallow oxygen minimum zone. To assess the influence of oxygen depleted eddies on the oxygen budget of the upper water column, a sub-region between the ventilation pathways of the subtropical gyre and the zonal current bands of the equatorial Atlantic was chosen and investigated in more detail. This region includes the most pronounced shallow oxygen minimum and is in the following referred to as shallow oxygen minimum zone (SOMZ, Fig. 1a). The probability of "dead-zone" eddy occurrence per year is more or less evenly distributed in the ETNA (Fig. 1a). Particularly in the SOMZ there seems to be neither a distinctly high nor an explicitly low "dead-zone" eddy occurrence. Due to the absence of other ventilation pathways in this zone, the influence of "dead-zone" eddies on the shallow oxygen minimum budget may be important and a closer examination worth the effort. We determine the average characteristics of "dead-zone" eddies in the ETNA, addressing their hydrographic features as well as occurrence, distribution, generation and frequency. Based on oxygen anomalies and eddy coverage we estimate their contribution to the oxygen budget of the SOMZ. The paper is organized as follows. Section 2 addresses the different in-situ measurements, satellite products and methods we use. Our results are presented in section 3, discussed in section 4 and summarized in section 5.

## 2. Data and methods

### 2.1 In-situ data acquisition

For our study we employ a quality-controlled database combining shipboard measurements, mooring data and Argo float profiles as well as autonomous glider data taken in the ETNA. For details on the structure and processing of the database see Schütte et al. (2016). For this study we extended the database in several ways. The region was expanded to now cover the region from 0° to 22° N and 13° W to 38° W (see Fig. 2). We then included data from five recent ship expeditions (RV Islandia ISL_00314, RV Meteor M105, M107, M116, M119), which sampled extensively within the survey region. Data from the two most recent deployment periods of the CVOO mooring from October 2012 to September 2015 as well as Argo float data for the years 2014 and 2015 were also included. Furthermore, oxygen measurements of all data sources were collected and integrated into the database. As the last modification of the database we included data from four autonomous gliders that were deployed in the region and sampled two ACMEs and one CE. Glider IFM11 (deployment ID: ifm11_depl01) was deployed on March 13, 2010. It covered the edge of an ACME on March 20 and recorded data in the upper 500 m. Glider IFM05 (deployment ID: ifm05_depl08) was deployed on June 13, 2013. It crossed a CE on July 26 and recorded data down to 1000 m depth. IFM12 (deployment ID: ifm12_depl02) was deployed on January 10, 2014 north of the Cape Verde island São Vicente and surveyed temperature, salinity and oxygen to 500 m depth. IFM13 (deployment ID: ifm13_depl01) was deployed on March 18, 2014 surveying temperature, salinity and oxygen to 700 m depth. IFM12 and IFM13 were able to sample three complete sections through an ACME. All glider data were internally recorded as a time series along the flight path, while for the analysis the data was interpolated onto a regular pressure grid of 1 dbar resolution (see also Thomsen et al., 2015). Gliders collect a large number of relatively closely spaced slanted profiles. To reduce the number of dependent measurements, we limited the number of glider profiles to one every 12 hours. All four autonomous gliders were equipped with Aanderaa optodes (3830) installed in the aft section of the devices. A recalibration of the Optode calibration coefficients were determined on dedicated CTD casts following the procedures of (Hahn et al., 2014). These procedures also estimates and correct the delays caused by the slow optode response time (more detailed information can be found in Hahn et al. (2014); Thomsen et al. (2015)). As gliders move through the water column the oxygen measurements are not as stable as those from moored optodes analyzed by Hahn et al. (2014). We thus estimate their measurement error to about 3 $\mu$mol kg$^{-1}$. The processing and quality control procedures for temperature and salinity data from shipboard measurements, mooring data and Argo floats has already been described by Schütte et al. (2016). The processing of the gliders' temperature and salinity measurements is described in Thomsen et al. (2015). Oxygen measurements of the shipboard surveys were collected with Seabird SBE 43 dissolved oxygen sensors attached to Seabird SBE 9plus or SBE 19 conductivity-temperature-depth (CTD) systems. Sampling and calibration followed the procedures detailed in the GO-SHIP manuals (Hood et al., 2010). The resulting measurement error were $\leq$1.5 $\mu$mol kg$^{-1}$. Within the CVOO moorings, a number of dissolved oxygen sensors (Aanderaa optodes type 3830) were used.

Calibration coefficients for moored optodes were determined on dedicated CTD casts and additional calibrated in the laboratory with water featuring 0% air saturation before deployment and after recovery following the procedures described by Hahn et al. (2014). We estimate their measurement error at <3 $\mu$mol kg$^{-1}$. For the few Argo floats equipped with oxygen sensors a full calibration is usually not available and only a visual inspection of the profiles was done before including the data into the database. The different manufacturers of Argo float oxygen sensors specify their measurement error at least better than 8 $\mu$mol kg$^{-1}$ or 5%, whichever is larger. Note

that early optodes can be significantly outside of this accuracy range, showing offsets of 15-20 µmol kg$^{-1}$, in
some cases even higher.
As a final result the assembled in-situ database of the ETNA contains 15059 independent profiles (Fig. 2). All
profiles include temperature, salinity and pressure measurements while 38.5% of all profiles include oxygen
measurements. The database is composed of 13% shipboard, 22.5% CVOO mooring, 63% Argo float and 1.5%
glider profiles. To determine the characteristics of different eddy types from the assembled profiles, we
separated them into CEs, ACMEs and the "surrounding area" not associated with eddy-like structures following
the approach of Schütte et al. (2016).
**2.2 Satellite data**
We detected and tracked eddies following the procedures described in Schütte et al. (2016). In brief we used 19
years of the delayed-time "all-sat-merged" reference dataset of SLA (version 2014). The data is produced by
Ssalto/Duacs and distributed by AVISO (Archiving, Validation, and Interpretation of Satellite Oceanographic),
with support from CNES [http://www.aviso.altimetry.fr/duac/]. We used the multi-mission product, which is
mapped on a 1/4° x 1/4° Cartesian grid and has a temporal resolution of one day. The anomalies were computed
with respect to a nineteen-year mean. The SLA and geostrophic velocity anomalies also provided by AVISO
were chosen for the time period January 1998 to December 2014.
For SST the dataset "Microwave Infrared Fusion Sea Surface Temperature" from Remote Sensing Systems
(www.remss.com) is used. It is a combination of all operational microwave (MW) radiometer SST
measurements (TMI, AMSR-E, AMSR2, WindSat) and infrared (IR) SST measurements (Terra MODIS, Aqua
MODIS). The dataset thus combines the advantages of the MW data (through-cloud capabilities) with the IR
data (high spatial resolution). The SST values are corrected using a diurnal model to create a foundation SST
that represents a 12-noon temperature (www.remss.com). Daily data with 9 km resolution from January 2002 to
December 2014 are considered.
For sea surface chlorophyll (Chl) data we use the MODIS/Aqua Level 3 product available at
http://oceancolor.gsfc.nasa.gov provided by the NASA. The data were measured via IR and is therefore cloud
cover dependent. Daily data mapped on a 4 km grid from January 2006 to December 2014 is selected.
**2.3 Low-oxygen eddy detection and surface composites**
In order to verify whether low oxygen concentrations (<40 µmol kg$^{-1}$) at shallow depth (above 200 m) are
associated with eddies we applied a two step procedure. First, all available oxygen measurements of the
combined in-situ datasets are used to identify negative oxygen anomalies with respect to the climatology. Next,
the satellite data based eddy detection results (Schütte et al., 2016) were matched in space and time with the
location of anomalously low-oxygen profiles. In this survey the locations of 173 of 180 low-oxygen profiles
coincide with surface signatures of mesoscale eddies. Schütte et al. (2016) showed that ACMEs can be
distinguished in the ETNA from "normal" anticyclonic eddies by considering the SST anomaly (cold in case of
ACMEs) and sea surface salinity (SSS) anomaly (fresh in case of ACMEs) in parallel to the respective SLA
anomaly. The satellite-based estimates of SLA and SST used in this study are obtained by subtracting low-pass
filtered (cutoff wavelength of 15° longitude and 5° latitude) values from the original data to exclude large-scale
variations and preserve only the mesoscale variability (see Schütte et al. (2016) for more detail). All eddy-like

structures with low-oxygen profiles are visually tracked in the filtered SLA (sometimes SST data) back- and forward in time in order to obtain eddy propagation trajectories. The surface composites of satellite-derived SLA, SST and Chl data consist of 150 km x 150 km snapshots around the obtained eddy centers. For construction of the composites the filtered SLA and SST is used as well.

**2.4 Reconstruction of oxygen concentrations in low-oxygen eddy cores**

About 30 % of the profiles from the combined in-situ dataset conducted in CEs or ACMEs do not have oxygen measurements available. However, we are only interested in oxygen measurements in isolated CE or ACME cores. These isolated eddy cores carry anomalously low salinity SACW of coastal origin, while the surrounding waters are characterized by an admixture of more saline NACW (Schütte et al., 2016) . All eddies that show a low salinity and cold core indicate that (I) they have been generated near the coast and (II) their core has been efficiently isolated from surrounding waters. The salinity-$\sigma_\theta$ diagram (Fig. 3a) of open ocean (west of 19°W) profiles shows a correlation between low salinity eddy cores and low oxygen concentrations. Moreover, it indicated that the oxygen content in the isolated eddies is decreasing from east to west. In order to compensate for missing oxygen measurements on many of the profiles we derive a salinity-oxygen relation but also considering the "age" of the eddy (time since the eddy left the eastern boundary) and an oxygen consumption rate within the eddy core. The oxygen consumption rate is estimated from the difference between the observed oxygen and a reference profile (the mean of all profiles east of 18°W in the eastern boundary region; Fig. 3a), the distance from the eastern boundary, and the propagation speed (3 km d$^{-1}$; see Schütte et al. (2016)). The mean eddy consumption rate is now the difference from the initial oxygen condition and the observed oxygen concentration in the eddy core divided by the eddy age (distance divided by propagation speed). For eddy profiles without oxygen measurements but SACW water mass characteristics (less saline and colder water than surrounding water) we can assume a strong isolation of the eddy and thus a lowering in oxygen. Using the coastal reference profile (Fig. 3), oxygen consumption rate and the distance from the coast an oxygen profile is reconstructed for all isolated CEs and ACMEs. To validate the method we reconstructed the oxygen profiles for the eddies with available oxygen measurements and compared them (Fig. 3b). On average an uncertainty of $\pm$ 12 (16) µmol kg$^{-1}$ is associated with the reconstructed oxygen values (Fig. 3c) of CEs (ACMEs). Depending on the intensity of isolation of the eddy core, lateral mixing could have taken place, which is assumed to be zero in our method. However, this approach enables us to enlarge the oxygen dataset by 30%. We considered the reconstructed oxygen profiles only to estimate the mean structure of oxygen anomaly.

**2.5 Mean vertical oxygen anomaly of low-oxygen eddies and their impact on the SOMZ**

To illustrate mean oxygen anomalies for CEs and ACMEs as a function of depth and radial distance, all oxygen profiles (observed and reconstructed) were sorted with respect to a normalized distance, which is defined as the actual distance of the profile from the eddy center divided by the radius of the eddy (the shape and thus the radius of the eddy are gained from the streamline with the strongest swirl velocity around a center of minimum geostrophic surface velocity). The oxygen profiles were grouped and averaged onto a grid of 0.1 increments between 0 and 1 of the normalized radial distance. Finally a running mean over three consecutive horizontal grid points was applied. A mean oxygen anomaly for the CEs and the ACMEs was constructed by the comparison with the oxygen concentrations in the surrounding waters. To illustrate the influence of the reconstructed oxygen values, the mean oxygen anomaly is also constructed based only on original measured oxygen values, both

anomalies are shown for comparison.
An oxygen deficit profile due to "dead-zone" eddies in the SOMZ is derived by building an oxygen anomaly on
density surfaces ($O_2'$) separating CEs and ACMEs. The derived anomalies are multiplied by the mean number of
eddies dissipating in the SOMZ per year ($n$) and weighted by the area of the eddy compared to the total area of
the SOMZ ($A_{SOMZ}$ = triangle in Fig. 1a). Differences in the mean isopycnal layer thickness of each eddy type
and the SOMZ are considered by multiplying the result with the ratio of the mean Brunt-Väisälä frequency ($N^2$)
outside and inside the eddy, resulting in an apparent oxygen utilization rate ($\mu$mol kg$^{-1}$ y$^{-1}$) due to "dead-zone"
eddies in the SOMZ on density layers:

$$aOUR = nO_2' \frac{\pi r_{Eddy}^2 N_{SOMZ}^2}{A_{SOMZ} N_{Eddy}^2}$$

where $r_{Eddy}$ is the mean radius of the eddies.

## 3. Results

### 3.1 Low-oxygen eddy observation from in-situ data

Several oxygen measurements in the ETNA with anomalously low oxygen concentrations, which is defined here as an oxygen concentration below 40 µmol kg$^{-1}$ (Stramma et al., 2009) could be identified from Argo floats, ship surveys, glider missions and from the CVOO mooring (Fig. 4). In total, 27 independent eddies with oxygen values <40 µmol kg$^{-1}$ in the upper 200 m were sampled with 173 profiles from 25 different platforms (Tab. 1). Almost all of the observed anomalous low oxygen values could be associated with mesoscale structures at the sea surface (CEs or ACMEs) from satellite data.

In-situ measurements for meridional velocity, temperature, salinity and oxygen of the CVOO mooring during the westward passage of one CE and one ACME with low oxygen concentrations are chosen to introduce the two different eddy types and their vertical structure based on temporally high resolution data (Fig. 5). From October 2006 to December 2006 (Fig. 5a), a CE passed the CVOO mooring position on a westward trajectory. At its closest, the eddy center was located about 20 km north of the mooring. The meridional velocities show a strong cyclonic rotation (first southward, later northward) with velocity maxima between the surface and 50 m depth at the edges of the eddy. In the core of the CE, the water mass was colder and less saline than the surrounding water, the mixed layer (ML) depth is reduced and the isopycnals are shifted upwards. The oxygen content of the eddy core was reduced by about 60 µmol kg$^{-1}$ at 115 m depth (or at the isopycnal surface 26.61 kg m$^{-3}$) compared to surrounding waters, which have a mean (± 1 standard deviation) oxygen content of 113 (± 38) µmol kg$^{-1}$ at around 150 m depth or 26.60 (± 0.32) kg m$^{-3}$ during the mooring period between 2006 to 2014. Schütte et al. (2016) showed that around 52% of the eddies in the ETNA represents CEs. They have a marginal smaller radius, rotate faster and have a shorter lifetime compared to the anticyclonic eddies, which is also shown in other observational studies of Chaigneau et al. (2009), Chelton et al. (2011), and theoretically suggested by Cushman-Roisin et al. (1990).

From January 2007 to March 2007 (Fig. 5b), an ACME passed the CVOO mooring position. The core of the westward propagating eddy passed about 13 km north of the mooring. The velocity field shows strong subsurface anticyclonic rotation at the depth of the core, i.e. between 80 to 100 m. In contrast to "normal" anticyclonic eddies, the water mass in the core of an ACME is colder and less saline than the surrounding waters. The isopycnals above the core are elevated resulting in shallower MLs both resembling a cyclone. Beneath the core, the isopycnals are strongly depressed as in a normal anticyclone. Thus, dynamically this resembles a mode water anticyclone, an eddy type, which is well-known from local single observations in almost all ocean basins (globally: Kostianoy and Belkin (1989); Mcwilliams (1985) "submesoscale coherent vortices (SCV)"; in the North Atlantic: Riser et al. (1986); Zenk et al. (1991) and Bower et al. (1995); Richardson et al. (1989); Armi and Zenk (1984) "Meddies"; in the Mediterranean Sea: Taupier‑Letage et al. (2003) "Leddies"; in the North Sea: Van Aken et al. (1987); in the Baltic Sea Zhurbas et al. (2004); in the Indian Ocean: Shapiro and Meschanov (1991) "Reddies"; in the North Pacific: Lukas and Santiago-Mandujano (2001); Molemaker et al. (2015) "Cuddies"; in the South Pacific: Stramma et al. (2013); Colas et al. (2012); Combes et al. (2015); Thomsen et al. (2016) and Nof et al. (2002) "Teddies"; in the Artic Dasaro (1988); Oliver et al. (2008)). For the majority of the observed mode-water type eddies the depressed isopycnals in deeper water mask the elevated isopycnals in the shallow water in terms of geostrophic velocity, resulting in an anticyclonic surface rotation, and a weak positive SLA (Gaube et al., 2014).

In contrast to most of the ACMEs reported the CVOO ACME eddy core is located at very shallow depth, just beneath the ML. The oxygen content in the eddy's core recorded from the CVOO mooring is strongly decreased with values around 19 µmol kg$^{-1}$ at 123 m depth (or 26.50 kg m$^{-3}$) compared to the surrounding waters (113 (± 38) µmol kg$^{-1}$). Within the entire time series, the CVOO mooring recorded the passage of several ACMEs with even lower oxygen concentrations (for more information see Karstensen et al. (2015) or Table 1). Recent model studies suggest that ACMEs represent a non-negligible part of the worlds eddy field, particular in upwelling regions (Combes et al., 2015; Nagai et al., 2015). Schütte et al. (2016) could show, based on observational data that ACMEs represent around 9% of the eddy field in the ETNA. Their radii are in the order of the first baroclinic mode Rossby radius of deformation and their eddy cores are well isolated (Schütte et al., 2016).

### 3.2 Combining in-situ and satellite data for low-oxygen eddy detection in the ETNA

Combining the location and time of in-situ detection of low-oxygen eddies with the corresponding SLA satellite data reveal a clear link to the surface manifestation of mesoscale structures, CE and ACMEs likewise (Fig. 4). Composite surface signatures for SLA, SST and Chl from all anomalous low-oxygen eddies as identified in the in-situ dataset are shown in Figure 6. The ACME composites are based on 17 independent eddies and on 922 surface maps. The detected ACMEs are characterized by an elevation of SLA, which is associated with an anticyclonic rotation at the sea surface. The magnitude of the SLA displacement is moderate compared to normal anticyclones and CEs (Schütte et al., 2016). More distinct differences to normal anticyclones are the cold-water anomaly and the elevated Chl concentrations in the eddy center of the ACMEs. Normal anticyclones are associated with elevated SST and reduced Chl concentrations. Through a combination of the different satellite products (SLA, SST, SSS) it is possible to determine low-oxygen eddies from satellite data alone (further details of the ACME tracking and the average satellite surface signatures (SLA, SST, SSS) of all eddy types (CEs, anticyclones and ACMEs) identified in 19 years of satellite data in Schütte et al. (2016)).

The composite mean surface signature for low-oxygen CEs is based on 10 independent eddies and on 755 surface maps. The CEs are characterized by a negative SLA and SST anomaly. The observed negative SST anomaly of the low-oxygen CEs is twice as large (core value CE: -0.12 (± 0.2) °C; core value ACME: -0.06 (± 0.2) °C) as the corresponding anomaly of the ACMEs. The Chl concentration in the eddy center is also higher for CEs compared to ACMEs (core value CE: 0.35 (± 0.22) log mg m$^{-3}$; core value ACME: 0.21 (± 0.17) log mg m$^{-3}$). Note, that we only considered the measured low-oxygen ACMEs and CEs from Table 1 to derive the composites.

Using the eddy-dependent surface signatures in SLA, SST and Chl the low-oxygen eddies could be tracked and an eddy trajectory could be derived (e.g. Fig. 4). All detected eddies were propagating westward into the open ocean. North of 12°N, most of the eddies set off near the coast, whereas south of 12°N the eddies seem to be generated in the open ocean. Detected CEs have a tendency to deflect poleward on their way into the open ocean (Chelton et al., 2011), whereas ACMEs seem to have no meridional deflection. However, during their westward propagation the oxygen concentration within the low-oxygen eddy cores decreases with time. Using the propagation time and an initial coastal oxygen profile (Fig. 3b) a mean apparent oxygen utilization rate per day could be derived for all sampled eddies (Fig. 7). On average the oxygen concentration decreases by about 0.19 ± 0.08 µmol kg$^{-1}$ d$^{-1}$ in the core of an isolated ACME, but has no significant trend in the core of an isolated CE (0.10 ± 0.12 µmol kg$^{-1}$ d$^{-1}$). This is in the range of recently published aOUR estimates for single observations of

CEs (Karstensen et al., 2015) and ACMEs (Fiedler et al., 2016).
**3.4 Mean oxygen anomalies from low-oxygen eddies in the ETNA**
In Figure 8 we compare the mean oxygen anomalies based purely on observations with those based on the
extended profile database including observed and reconstructed oxygen values (see section 2.4). It shows the
mean oxygen anomalies against the surrounding water for CE (Fig. 8a) and ACME (Fig. 8b) versus depth and
normalized radial distance. On the left side of each panel the anomaly is based on the observed and reconstructed
oxygen values (736 oxygen profiles; 575 in CEs; 161 in ACMEs), whereas on the right side the anomaly is based
only on the observed oxygen measurements (504 oxygen profiles; 395 in CEs; 109 in ACMEs). The distinct
mean negative oxygen anomalies for CEs and ACMEs indicate the low oxygen concentrations in the core of
both eddy types compared to the surrounding water. The strongest oxygen anomalies are located in the upper
water column, just beneath the ML. CEs feature maximum negative anomalies of around -100 $\mu$mol kg$^{-1}$ at
around 70 m depth in the eddy core, with a slightly more pronounced oxygen anomaly when including the
reconstructed values (left side of Fig.8) compared to the oxygen anomaly based purely on observation (right side
of Fig. 8a). This is contrary for the ACME with stronger oxygen anomalies on the right part than on the left (Fig.
8b). Both methods deliver maximum negative anomalies of around -120 $\mu$mol kg$^{-1}$ at around 100 m depth in the
ACME core. At that depth, the diameter of the mean oxygen anomaly is about 100 km for ACMEs and 70 km
for CEs (the eddy core is defined here as the area of oxygen anomalies smaller than -40 $\mu$mol kg$^{-1}$). Beneath 150
m depth, magnitude and diameter of the oxygen anomalies decrease rapidly for both eddy types. Figure 8c is
based on both, the in-situ and reconstructed oxygen values, and shows the horizontal mean oxygen anomaly
profile of each eddy type against depth obtained by horizontally averaging the oxygen anomalies shown in Fig.
8a,b. The maximum anomalies are -100 $\mu$mol kg$^{-1}$ at around 90 m for ACMEs and -55 $\mu$mol kg$^{-1}$ at around 70 m
for cyclones. Both eddy types have the highest oxygen variance directly beneath the ML (in the eddy core) or
slightly above the eddy core. The oxygen anomaly (and associated variance) decreases rapidly with depth
beneath the eddy core and is smaller than around -10 ± 10 $\mu$mol kg$^{-1}$ beneath 350 m for both eddy types.
**4. Discussion**
The pelagic zones of the ETNA are traditionally considered to be "hypoxic", with minimal oxygen
concentrations of marginally below 40 $\mu$mol kg$^{-1}$ (Brandt et al., 2015; Karstensen et al., 2008; Stramma et al.,
2009). This is also true for the upper 200 m (Fig. 1). However, single oxygen profiles taken from various
observing platforms (ships, moorings, gliders, floats) with oxygen concentrations in the range of severe hypoxia
(< 20 $\mu$mol kg$^{-1}$) and even anoxia (~ 1 $\mu$mol kg$^{-1}$) conditions and consequently below the canonical value of 40
$\mu$mol kg$^{-1}$ (Stramma et al., 2008) are found in a surprisingly high number (in total 180 profiles) in the ETNA. In
the current analysis we could associate observations of low-oxygen profiles with 27 independent mesoscale
eddies (10 CEs and 17 ACMEs). Mesoscale eddies are defined as coherent, nonlinear structures with a lifetime
of several weeks to more than a year and radii larger than the first baroclinic mode Rossby radius of deformation
(Chelton et al., 2007). In reference to the surrounding water, the eddies carry a negative oxygen anomaly which
is most pronounced right beneath the mixed layer. The oxygen anomaly is attributed to both, an elevated primary
production in the surface layers of the eddies (documented by positive chlorophyll anomalies estimated from
satellite observations, Fig. 6) and the subsequent respiration of organic material (Fiedler et al., 2016), and the
dynamically induced isolation of the eddies with respect to lateral oxygen resupply (Fiedler et al., 2016;
Karstensen et al., 2015). In contrast to the transport of heat or salt with ocean eddies the oxygen anomaly
intensified with time the eddy exists (eddy age). The oxygen depleted eddy cores are either associated to CEs or
ACMEs. In the ETNA both eddy types have in common that in their center the mixed layer base rises towards
shallow depth (50 to 100m) which in turn favor biological productivity in the euphotic zone (Falkowski et al.,
1991; McGillicuddy et al., 1998). In addition, an enhanced vertical flux of nutrients within or at the periphery of
the eddies due to submesoscale instabilities is expected to occur (Brannigan et al., 2015; Karstensen et al., 2016;
Lévy et al., 2012; Martin and Richards, 2001; Omand et al., 2015).
As a consequence the eddies establish an specific ecosystem of high primary production, particle load and
degradation processes, and even unexpected  nitrogen loss processes (Löscher et al., 2015). The combination of
high productivity and low oxygen supply resample the process of "dead zone" formation, know from other
aquatic systems. As for other aquatic systems specific threats to the ecosystem of the eddies are observed such as
the interruption of the diurnal migration of zooplankters (Hauss et al., 2016).
We observed low-oxygen cores only in ACMEs (also known as "submesoscale coherent vortices (SCV)"
(Dasaro, 1988; Mcwilliams, 1985) or "intra-thermocline eddies" (Kostianoy and Belkin, 1989)) and CEs but not
in normal anticyclonic rotating eddies. In fact the mixed layer base in normal anticyclonic eddies is deeper than
the surroundings, bending downward towards the eddy center as a consequence of the anticyclonic rotation.
Therefore the normal anticyclones create a positive oxygen anomalies when using depth levels as a reference.
However, when using density surfaces as a reference the anomalies disappear. Moreover, normal anticyclonic
eddies have been found to transport warm and salty anomalies (Schütte et al., 2016) along with the positive
oxygen anomaly which is very different from the ACMEs (and CEs) with a low-oxygen core.
The ETNA is expected to have a rather low population of long-lived eddies (Chaigneau et al., 2009; Chelton et
al., 2011), we could identify 234 CEs and 18 ACMEs per year in the ETNA with a radius > 45 km and a tracking
time of more than 3 weeks. For the eddy detection we used an algorithm based on the combination of the Okubo-
Weiß method and a modified version of the geometric approach from (Nencioli et al. (2010)) with an adjusted
tracking for the ETNA (for more information see Schütte et al. (2016)). Schütte et al. (2016) found an eddy-type
depended connection between SLA and SST (and SSS) signatures for the ETNA that allowed a detection (and
subsequently closer examination) of ACMEs. Because of weaker SLA signatures, the tracking of ACMEs is
rather difficult due to the small signal to noise ratio (not the case for the CEs) and automatic tracking algorithms
may fail in many cases. Note, all tracks of ACMEs and CEs shown in Figure 4 were visually verified. Similar to
what Schütte et al. (2016), did we derived "dead-zone" eddies surface composites for SST, SSS (not shown here)
and Chl (Fig. 6). It revealed that the existence of an ACMEs is very associated with low SST (and SSS) but also
with high Chl (see also single maps in Karstensen et al. 2015). Analyzing jointly SLA, SST and Chl maps we
found that ACMEs represent a non-negligible part of the eddy field (32% normal anticyclones, 52% CEs, 9%
ACMEs (Schütte et al., 2016)).
It has been shown (Fig. 4) that the low-oxygen eddies in the ETNA could be separated into two different
regimes, north and south of 12°N. The eddies north of 12°N are generally generated along the coast and in
particular close to the headlands along the coast. Schütte et al. (2016) suggested that CEs and normal
anticyclones north of 12°N are mainly generated from instabilities of the northward directed alongshore
Mauretania Current (MC), whereas the ACMEs are most likely generated by instabilities the Poleward
Undercurrent (PUC). However, the detailed generation processes need to be further investigated. The low-
oxygen eddies south of 12°N do not originate from a coastal boundary upwelling system. Following the

trajectories it seems that the eddies are generated in the open ocean between 5°N and 7°N. In general, the occurrence of oxygen depleted eddies south of 12°N is rather astonishing, as due to the smaller Coriolis parameter closer to the equator the southern eddies should be more short-lived and less isolated compared to eddies further north. In addition, the generation mechanism of the southern eddies is not obvious. The eddy generation could be related to the presence of strong tropical instabilities in that region (Menkes et al., 2002; von Schuckmann et al., 2008). However, in particular the generation of ACMEs is complex and has been subject of scientific interest for several decades already (Dasaro, 1988; Mcwilliams, 1985). The low stratification of the eddy core cannot be explained by pure adiabatic vortex stretching alone as this mechanism will result in cyclonic vorticity, assuming that $f$ dominates the relative vorticity. Accordingly, the low stratification in the eddy core must be the result of some kind of preconditioning induced by for example upwelling, deep convection (Oliver et al., 2008) or diapycnal mixing near the surface or close to boundaries (Dasaro, 1988) before eddy generation takes place (Mcwilliams, 1985). Dasaro (1988), Molemaker et al. (2015) and Thomsen et al. (2015) highlight the importance of flow separation associated with headlands and sharp topographical variations for the generation of ACMEs. This notion is supported by the fact that low potential vorticity signals are usually observed in the ACMEs (Dasaro, 1988; Mcwilliams, 1985; Molemaker et al., 2015; Thomas, 2008). The low potential vorticity values suggest that the eddy has been generated near the coast as - at least in the tropical latitudes - such low potential vorticity values are rarely observed in the open ocean. These theories seem to be well suitable for the ACME generation north of 12°N but do not entirely explain the occurrence of ACMEs south of 12°N. However, more research on this topic is required.

Because we expect "northern" and "southern" eddies to have different generation mechanisms and locations and because they have different characteristics we discuss them separately. The core of the eddies generated north of 12°N is characterized by less saline and cold SACW (Schütte et al., 2016) and thereby forms a strong hydrographic anomaly against the background field. On the contrary, the core of the eddies generated south of 12°N does not show any significant hydrographic anomalies. However, given the low-oxygen core in eddies in both regions we expect that the processes that create the "dead-zone", which is isolation and high productivity, are also present in both regimes. The oxygen content decrease on average by about $0.19 \pm 0.08$ µmol kg$^{-1}$ d$^{-1}$ in an ACME and by about $0.10 \pm 0.12$ µmol kg$^{-1}$ d$^{-1}$ in an CE, based on 504 oxygen measurements in CEs and ACMEs. Note, that these apparent oxygen utilization rates (aOUR) are in the range of recently published aOUR estimates for CEs (Karstensen et al., 2015) and ACMEs (Fiedler et al., 2016), which are based on single measurements in "dead-zone" eddies. In particular for CEs we take that as an indication that no significant trend in aOUR exists. An important point regarding the method and the associated inaccuracies in deriving the aOURs is the initial coastal oxygen concentration, which is highly variable in coastal upwelling regions (Thomsen et al., 2015). In addition one should mention that the relative magnitude of eddy dependent vertical nutrient flux, primary productivity and associated oxygen consumption or nitrogen fixation/denitrification in the eddy cores strongly varies between different eddies, because of differences in the initial water mass in the eddies' core, the eddies' age and isolation and the experienced external forcing (in particular wind stress and dust/iron input).

However, the mean oxygen profiles from the eastern boundary and inside of all CEs and ACMEs (Fig. 3b) indicate no pronounced oxygen difference beneath 250 m depth. The largest anomalies have been observed in the eddy cores at around 100 m depth (Fig. 8). As a result of the dynamic structure, the core water mass anomalies of the ACMEs are more pronounced than the one of the CE (Karstensen et al., 2016) and consequently the oxygen anomalies are stronger. This is supported by the differences in the oxygen anomaly

based on the measured plus reconstructed and the measured oxygen values. The reconstruction of oxygen values assumes a complete isolation of the eddy core. The left side of Figure 8a, which includes the reconstructed oxygen values, features a larger oxygen anomaly than the right side based on measured oxygen values only. Consequently the CEs are probably not completely isolated and the evolving oxygen anomaly is affected by some lateral flux of oxygen. On the contrary, the oxygen anomaly of ACMEs (Fig. 8b) is smaller for the reconstruction than for the measured oxygen values. This suggest that the ACMEs are more effectively isolated resulting in enhanced apparent consumption in the ACME core. However, another source of error in the reconstructed oxygen values is the assumption of a linear decrease of oxygen with time. All observed CEs or ACMEs contain a negative oxygen anomaly, partly because they transport water with initial low oxygen concentrations and additionally because the oxygen consumption in the eddies is more intense then in the surrounding waters (Karstensen et al. 2015, Fiedler et al. 2016). Dasaro (1988), Molemaker et al. (2015) and Thomsen et al. (2015) argued that the core waters of ACME's generated near the coast originate to a large extent from the bottom boundary layer at the continental slopes. At the shelf off Northwest Africa occasionally low oxygen concentrations (around 30 µmol kg$^{-1}$) in the depth range between 50 to 150 m could locally identified (M. Dengler personal communication). Consequently it is certainly possible that the eddies have initially low oxygen concentrations in their cores. This is not the case for the short-lived southern eddies, which seem to be generated in the open ocean. It would suggest that, to achieve similarly strong negative oxygen anomalies, the oxygen consumption in the eddies south of 12°N must be even stronger than in the ACMEs further north. Pronounced productivity patterns in tropical instability waves and vortices have been reported in the past (Menkes et al., 2002), but were not connected to low-oxygen eddies before.

In the following, an estimate of the contribution of the negative oxygen anomalies of low-oxygen eddies to the oxygen distribution of the SOMZ is presented. The satellite-based eddy tracking reveals that on average each year 14 (2) CEs (ACMEs) are propagating from the upwelling system near the coast into the SOMZ and dissipate there. By deriving the oxygen anomaly on density surfaces an oxygen loss profile due to low-oxygen eddies in the SOMZ is derived (Fig. 9). Note that due to the lower oxygen values within the eddies compared to the surrounding waters in the SOMZ, the release of negative oxygen anomalies to the surrounding waters is equivalent to a local (eddy volume) enhancement of the oxygen utilization by -7.4 (-2.4) µmol kg$^{-1}$ yr$^{-1}$ for CEs (ACMEs) for the depth range of the shallow oxygen minimum in the SOMZ, i.e. 50 to 150 m depth. Instead of describing the effect of the low-oxygen eddies on the oxygen consumption an equivalent view is to consider a box model approach for the SOMZ. The basis of this box model is the mixing of high-oxygen waters (the background conditions) with low-oxygen waters (the low-oxygen eddies). The average oxygen concentrations within the eddies in the considered depth range, i.e. 50 to 150 m depth, are 73 (66) µmol kg$^{-1}$ for CEs (ACMEs). The average oxygen concentration of the background field averaged over the same depth range (between 50 and 150 m depth) derived from the MIMOC climatology (Schmidtko et al., 2013) is 118 µmol kg$^{-1}$. This climatological value includes the contribution of low-oxygen eddies. If we now consider the respective oxygen concentrations and volumes of the SOMZ and the eddies (multiplied by their frequency of occurrence per year), we are able to calculate the theoretical background oxygen concentration for the SOMZ without eddies to be 125 µmol kg$^{-1}$. Naturally due to the dispersion of negative oxygen anomalies, the oxygen concentrations in the SOMZ without eddies must be higher than the observed climatological values. Attributing the difference of these oxygen concentrations on the one hand in the SOMZ without eddies (125 µmol kg$^{-1}$) and on the other hand the

observed climatological values in the SOMZ with eddies (118 µmol kg$^{-1}$), solely to the decrease induced by the dispersion of eddies, we find that an equivalent reduction of around 7 µmol kg$^{-1}$ of the observed climatological oxygen concentration in the SOMZ box. To visualize that a depth profile of oxygen in the SOMZ without the dispersion of low-oxygen eddies is equally derived and compared to the observed oxygen profile in the SOMZ (Fig. 9b). Consequently, the oxygen consumption in this region is a mixture of the large-scale metabolism in the open ocean (Karstensen et al. 2008) and the enhanced metabolism in low-oxygen eddies (Karstensen et al. 2016, Fiedler et al. 2016). Note, that a small compensating effect for example due to diapycnal oxygen fluxes in normal anticyclones can probably be expected. However, our estimates should be considered as a lower limit for the contribution of AMCEs because of the problem in detecting and tracking ACMEs (weak SLA anomaly) and because of the assumption of zero lateral ventilation within the eddies. Moreover, we identified a few occurrences of ACMEs based on shipboard ADCP as well as hydrographic measurements (e.g. during the research cruises of Ron Brown 2009 and Meteor 119) that did not have a significant SLA signature. In addition only eddies are considered which could be followed with tracking algorithms directly from the coast into the transition zone and having a radius greater than 45 km and a lifetime of more than 21 days.

Although a reduction of 7 µmol kg$^{-1}$ seems to be small number one may note that the peak difference is a reduction of 16 µmol kg$^{-1}$ at 100 m depth (Fig. 9), in the core depth of the shallow oxygen minimum zone in the ETNA. The additional respiration due to the presence of low-oxygen eddies can be important as well in numerical simulations, where up to now only the large scale consumption is taken into account. In turn it is important to investigate the eddy occurrence and eddy cycling in numerical simulation of the OMZs given they have a sufficient resolution.

Our results question the assumption that the oxygen consumption is determined by the metabolism of the large-scale community alone. The observations presented here suggest instead that also hot spots of locally enhanced consumption may possibly need to be considered in the future.

## 5. Conclusion

In this study, we investigated the vertical structure of oxygen depleted eddies in the ETNA based on satellite (a combination of SLA and SST) and in-situ oxygen and hydrography data (ship data, mooring data, profiling floats, underwater glider). We frequently detected oxygen concentrations below the canonical value of 40 µmol kg$^{-1}$ within the ETNA that are associated with CEs and ACMEs. Lowest oxygen concentration in these eddies was observed at shallow depth, just underneath the mixed layer between 50 to 150 m. Both, CEs and ACMEs, are characterized by a positive Chl anomaly suggesting enhanced productivity in the eddy surface water. Respiration of the organic material, in combination with sluggish lateral oxygen fluxes across the eddy boundaries, most likely create the low-oxygen core. A process that resamples the creation of "dead-zones" but in the open ocean (Karstensen et al., 2015). Oxygen concentrations are found to decrease in the eddy cores during the westward propagation from their generation region along the West African coast into the open ocean. Our assessment reveals that 234 CEs (18 ACMEs) are generated each year (mostly on the eastern boundary) in the ETNA and can be tracked longer than 3 weeks (considered here as the time scale for coherent eddies). On average the oxygen concentration in the core of coherent CEs (ACMEs) decreases by about 0.10 (0.19) ± 0.12 (0.08) µmol kg$^{-1}$ d$^{-1}$. Beside the eddies originating in generation regions along the West African coast, we observe low-oxygen eddies (primarily ACMEs) relatively close to the equator, south of 12°N. These eddies may

be generated from flow instability processes occurring during the formation of tropical instability waves.
However, both types of eddies (north of 12°N and south of 12°N) contain their minimum oxygen concentration
in the depth range where a shallow oxygen minimum is found in the ETNA. A simple box model approach on
the basis of mixing ratios of high-oxygen waters with low-oxygen waters in the SOMZ reveals that a mean
reduction of around 7 µmol kg$^{-1}$ (peak reduction is 16 µmol kg$^{-1}$ at 100 m depth) of the observed oxygen in the
shallow oxygen minimum zone is explainable due to the dispersion of low-oxygen eddies. This value, though, is
very likely underestimated due to difficulties in identifying and tracking of ACMEs. The additional consumption
within these low-oxygen eddies represents a substantial part of the total consumption in the open ETNA and
might be partly responsible for the formation and extend of the shallow oxygen minimum. Given the impact of
ACMEs on the oxygen budget in the ETNA, a further distinction into the two types of anticyclonic eddies in
global (Chelton et al., 2011; Zhang et al., 2013) as well as regional eddy assessments is necessary, particular in
eastern boundary upwelling systems.
**Data availability**
The used satellite data SLA, SST and Chl can be freely downloaded at
http://www.aviso.altimetry.fr/en/data/products, http://www.remss.com/measurements/sea-surface-temperature/
and http://oceancolor.gsfc.nasa.gov, respectively. The Argo float data is freely available at
http://www.argodatamgt.org/Access-to-data/Argo-data-selection and the assembled shipboard measurements;
shipboard CTD, glider and CVOO mooring data used in this paper are available at
https://doi.pangaea.de/10.1594/PANGAEA.860778.
**Acknowledgements**
This study was funded by the Deutsche Bundesministerium für Bildung und Forschung (BMBF) as part of the
project AWA (01DG12073E), by the Deutsche Forschungsgemeinschaft through the Collaborative Research
Centre "SFB 754" and several research cruises with RV Meteor, RV Maria S. Merian, Ronald H. Brown and RV
L'Atalante. Furthermore by the Cluster of Excellence "The Future Ocean" (CP1341), the project "Eddy-Hunt"
(CP1341) and the BMBF project SOPRAN (03F0611A and 03F0662A). The CVOO mooring is part of the
OceanSITES mooring network. The captains and the crew as well as all chief scientists and scientists of the
research vessels and our technical group for their help with the fieldwork deserve special thanks. Furthermore
the authors thank Tim Fischer for continuing support and discussion and Rebecca Hummels for proof reading
and for assisting in improving this paper.
The Argo data using in this study were collected and made freely available by the International Argo Program
and the national programs that contribute to it. (http://www.argo.ucsd.edu, htpp://argo.jcommops.org). The Argo
Program is part of the Global Ocean Observing System. The Ssalto/Duacs altimeterproducts were produced and
distributed by the Copernicus Marine and Environment Monitoring Service (CMEMS)
(http://www.marine.copernicus.eu). The Microwave OI SST data are produced by Remote Sensing Systems and
sponsored by National Oceanographic Partnership Program (NOPP), the NASA Earth Science Physical
Oceanography Program, and the NASA MEaSUREs DISCOVER Project. Data are available at www.remss.com.
The chlorophyll_a version 6 is a remote dataset from the NASA Ocean Biology Processing Group (OBPG). The

1   OBPG is the official NASA data center that archives and distributes ocean color data

2   (http://oceancolor.gsfc.nasa.gov).

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

4 which are not included in Fig. 4 due to not existent delayed time satellite products.

| | Time | min O$_2$ between 0-200 m | Associated eddy type |
|---|---|---|---|
| **11 Ship-Cruises:** <br> **(81 profiles)** | | | |
| Meteor 68/3 | Summer 2006 | 17 | CE |
| L'Atalante GEOMAR 3 | Winter 2008 | 25 | ACME |
| Meteor 80/2 | Winter 2009 | 32 | ACME |
| Meteor 83/1 | Winter 2010 | 20 | ACME |
| Meteor 96 | Spring 2013 | 38 | ACME |
| Meteor 97 | Summer 2013 | 28 | ACME |
| Islandia | Spring 2014 | 10 | ACME |
| Meteor 105 | Spring 2014 | 4 | ACME |
| Meteor 116 | Spring 2015 | 17 | ACME* |
| Meteor 119 | Autumn 2015 | 30 | ACME* |
| Maria S. Merian 49 | Winter 2015 | 35 | CE* |
| **9 Argo floats:** <br> **(24 profiles)** | | | |
| 6900632 | Autumn 2008 | 14 | CE |
| 1900652 | Winter 2008 | 26 | ACME |
| 1900650 | Summer 2009 | 27 | ACME |
| 1901360 | Autumn 2014 | 34 | CE |
| 1901361 | Autumn 2014 | 21 | CE |
| 1901362 | Autumn 2014 | 26 | CE |
| 1901363 | Autumn 2014 | 37 | CE |
| 1901364 | Autumn 2014 | 24 | ACME |
| 1901365 | Autumn 2014 | 24 | ACME |
| **4 Gliders:** <br> **(32 profiles)** | | | |
| IFM 11 | Spring 2010 | 19 | ACME |
| IFM 05 | Summer 2013 | 9 | CE |
| IFM 12 | Winter 2014 | 1 | ACME |
| IFM 13 | Spring 2014 | 1 | ACME |
| **9 CVOO events:** <br> **(36 profiles)** | | | |
| Optode at 127 m depth | Winter 2007 | 15 | ACME |
| Optode at 79 m depth | Autumn 2008 | 38 | CE |
| Optode at 54 m depth | Winter 2010 | 2 | ACME |
| Optode at 53 m depth | Winter 2012 | 17 | ACME |

| Optode at 53 m depth | Spring 2012 | 30 | CE |
|---|---|---|---|
| Optode at 45 m depth | Summer 2013 | 29 | ACME |
| Optode at 45 m depth | Winter 2013 | 9 | CE |
| Optode at 43 m depth | Winter 2015 | 2 | ACME* |
| Optode at 43 m depth | Summer 2015 | 6 | ACME* |
| **∑ 173 profiles** | | | **∑ 27 different eddies** |

**Figures**

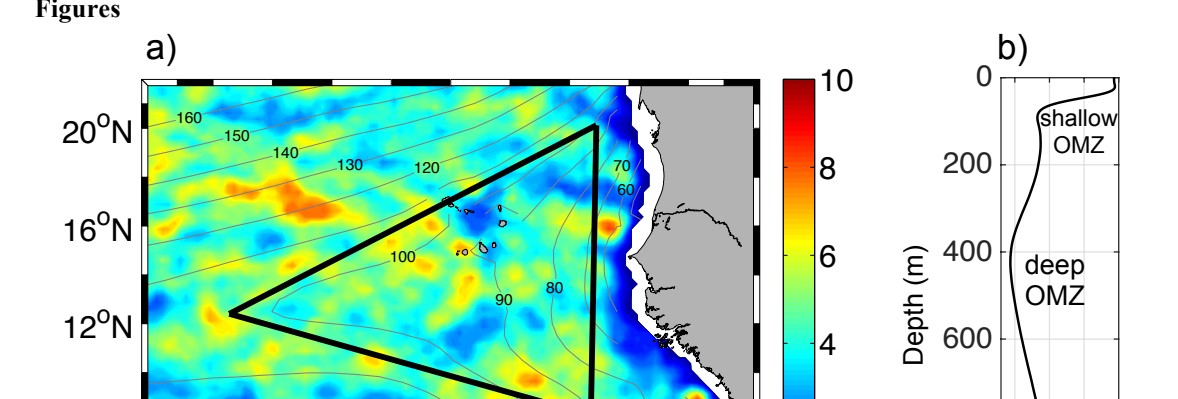

**Figure 1: a)** Map of the ETNA including contour lines of the oxygen minimum of the upper 200 m (in μmol kg⁻
¹) as obtained from the MIMOC climatology (Schmidtko et al., 2013). The color indicates the percentage of
"dead-zone" eddy coverage per year. The black triangle defines the SOMZ. **b)** mean vertical oxygen profile of
all profiles within the SOMZ showing the shallow oxygen minimum centered around 80 m depth and the deep
oxygen minimum centered at 450 m depth.

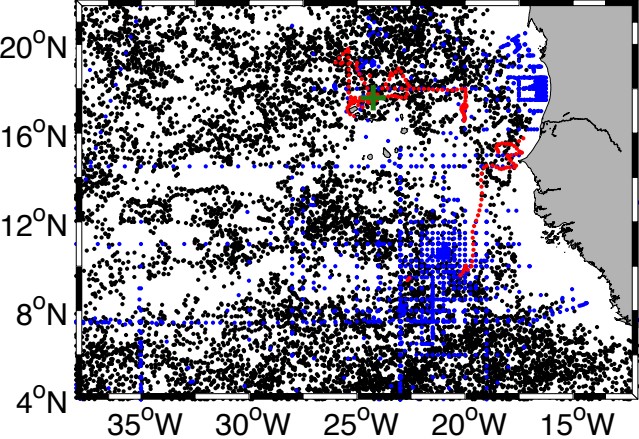

3  **Figure 2:** Map of the ETNA containing all available profiles between 1998 and 2014. The green cross marks the

4  CVOO position, blue dots mark shipboard CTD stations, red dots mark the locations of glider profiles and black

5  dots locations of Argo float profiles.

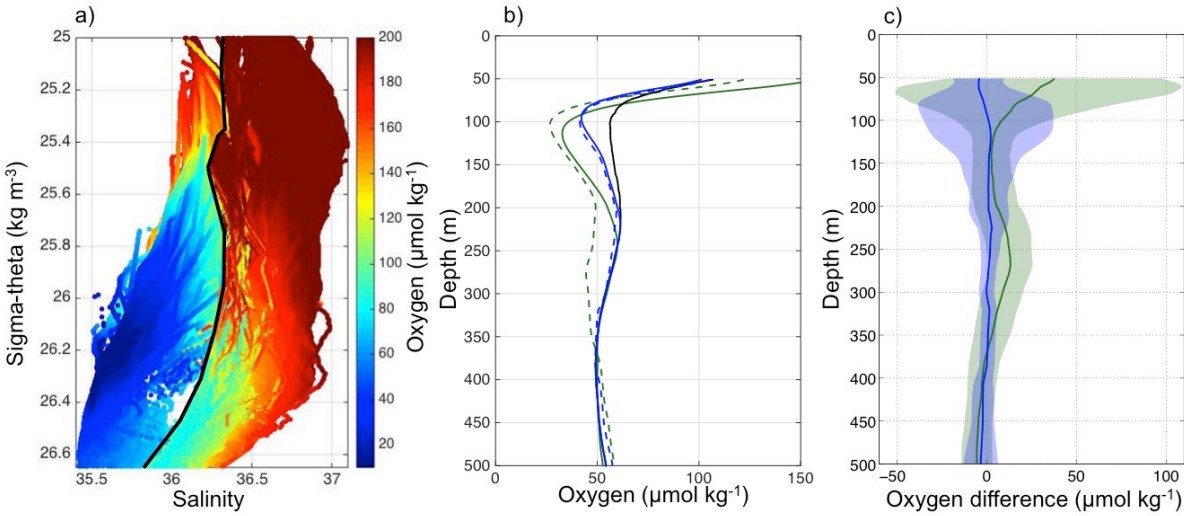

**Figure 3: a)** Salinity-$\sigma_0$ diagram with color indicating the oxygen concentrations. The black line separates the 173 profiles with minimum oxygen concentration of <40 µmol kg$^{-1}$ (left side / more SACW characteristics) from profiles of the surrounding water (right side / more NACW characteristics), taken from the same devices shortly before and after the encounter with a low-oxygen eddy. **b)** Mean oxygen concentration versus depth of the coastal region (east of 18°W, solid black line), of all CEs (solid blue line) and all ACMEs (solid green line) with available oxygen measurements. The dashed line represents the reconstructed mean oxygen concentration for the same CEs (blue) and ACMEs (green). **c)** Difference between the reconstructed and measured oxygen concentrations in CEs (blue) and ACMEs (green) with associated standard deviation (shaded area).

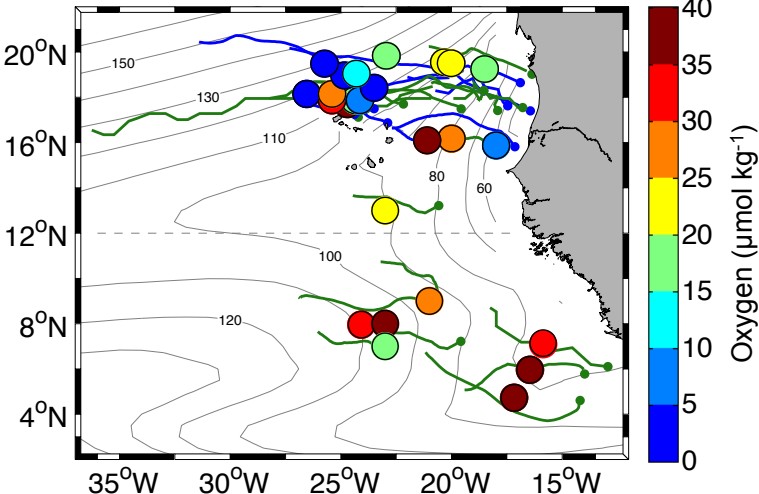

**Figure 4:** Minimum oxygen concentration (contour lines, µmol kg$^{-1}$) in the ETNA between the surface and 200 m depth as obtained from the MIMOC climatology (Schmidtko et al., 2013). Superimposed colored dots are all low-oxygen measurements (below 40 µmol kg$^{-1}$ in the upper 200 m) which could be associated with eddy-like structures. The size of the dots represents a typical size of the mesoscale eddies. The associated trajectories of the eddies are shown in green for ACMEs and in blue for cyclones. The oxygen concentrations are from the combined dataset of shipboard, mooring, glider and Argo float measurements.

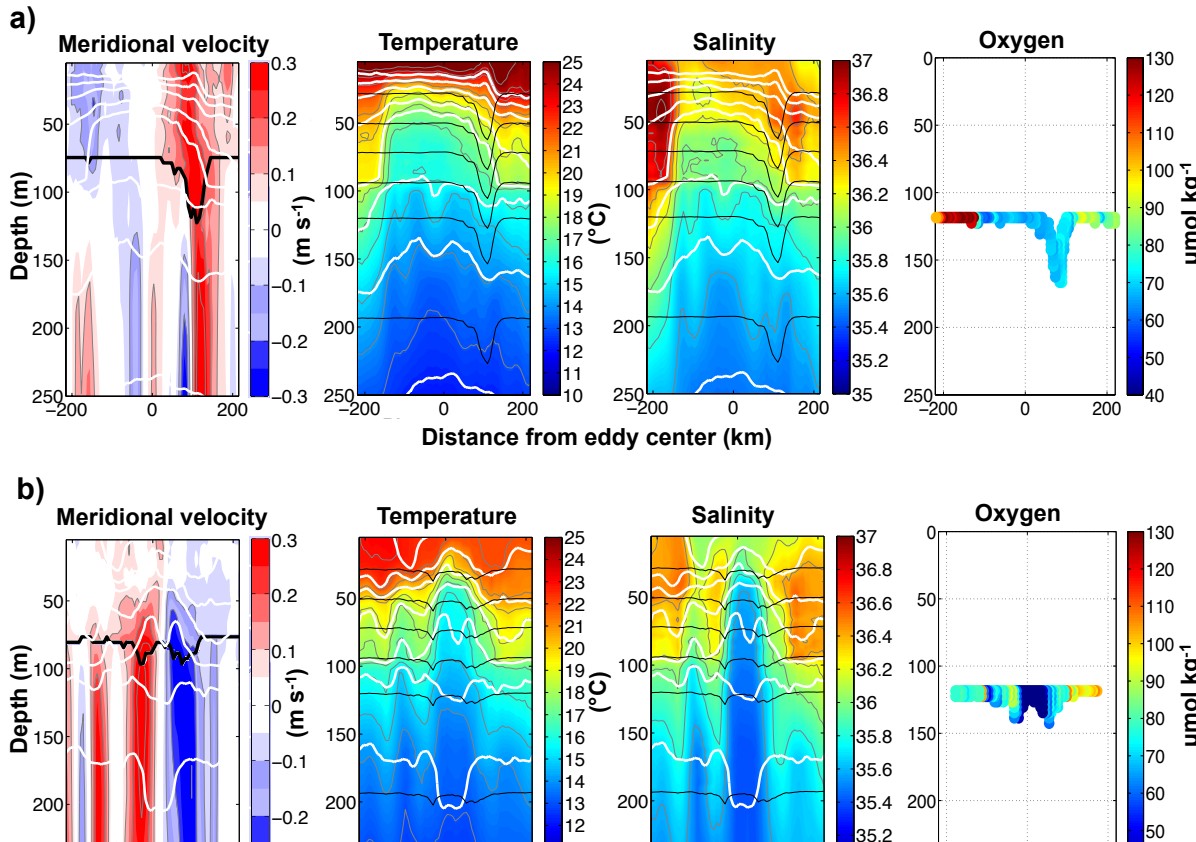

**Figure 5:** Meridional velocity, temperature, salinity and oxygen of an exemplary **a)** CE and **b)** ACME at the
CVOO mooring. Both eddies passed the CVOO on a westward trajectory with the eddy center north of the
mooring position (CE 20 km, ACME 13 km). The CE passed the CVOO from October to December 2006 and
the ACME between January and March 2007. The thick black lines in the velocity plots indicate the position of
an upward looking ADCP. Below that depth calculated geostrophic velocity is shown. The white lines represent
density surfaces inside the eddies and the thin grey lines isolines of temperature and salinity, respectively. Thin
black lines in the temperature and salinity plot mark the vertical position of the measuring devices. On the right
time series of oxygen is shown from the one sensor available at nominal 120 m depth.

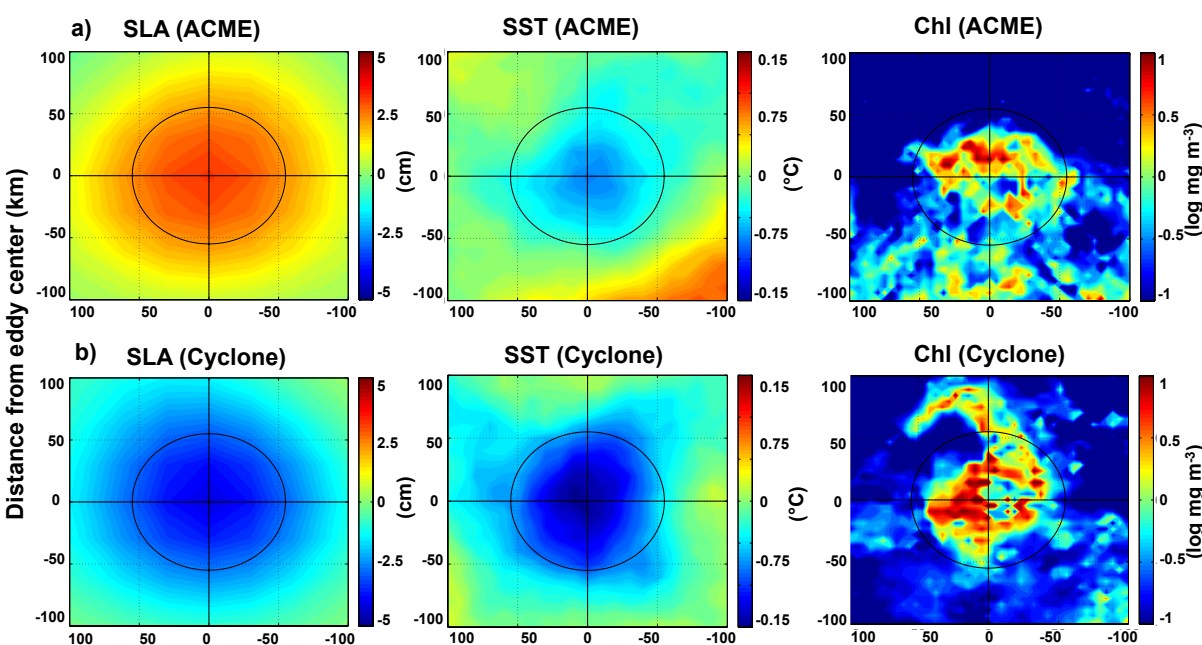

**Figure 6:** Composites of surface signature for SLA, SST and Chl from all detected low-oxygen eddies: **a)** ACMEs and **b)** CEs. The solid black cross marks the eddy center and the solid black circle the average radius. Due to significant cloud cover the number of Chl data are much less when compared to the SLA and SST data, thus there is more lateral structure.

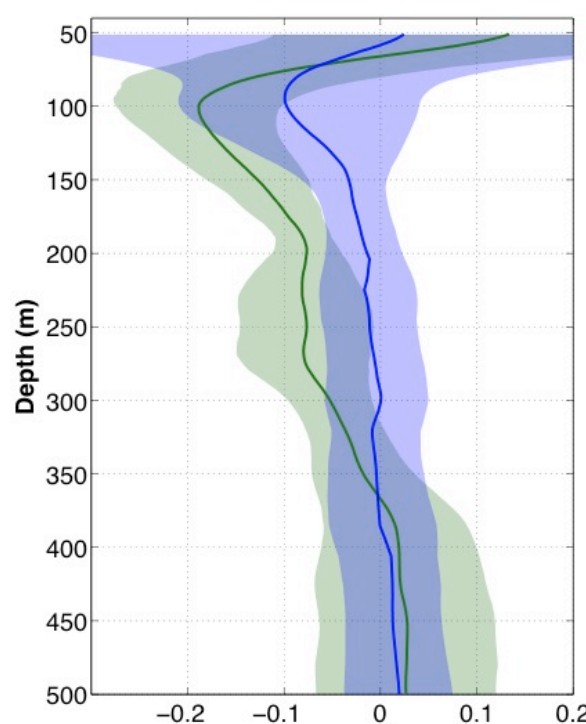

**Figure 7:** Depth profiles of a mean apparent oxygen utilization rate (aOUR, μmol kg$^{-1}$ d$^{-1}$) within CEs (blue) and
ACMEs (green) in the ETNA with associated standard deviation (shaded area). Derived by using the propagation
time of each eddy, an initial coastal oxygen profile and the assumption of linear oxygen consumption (based on
depth layers).

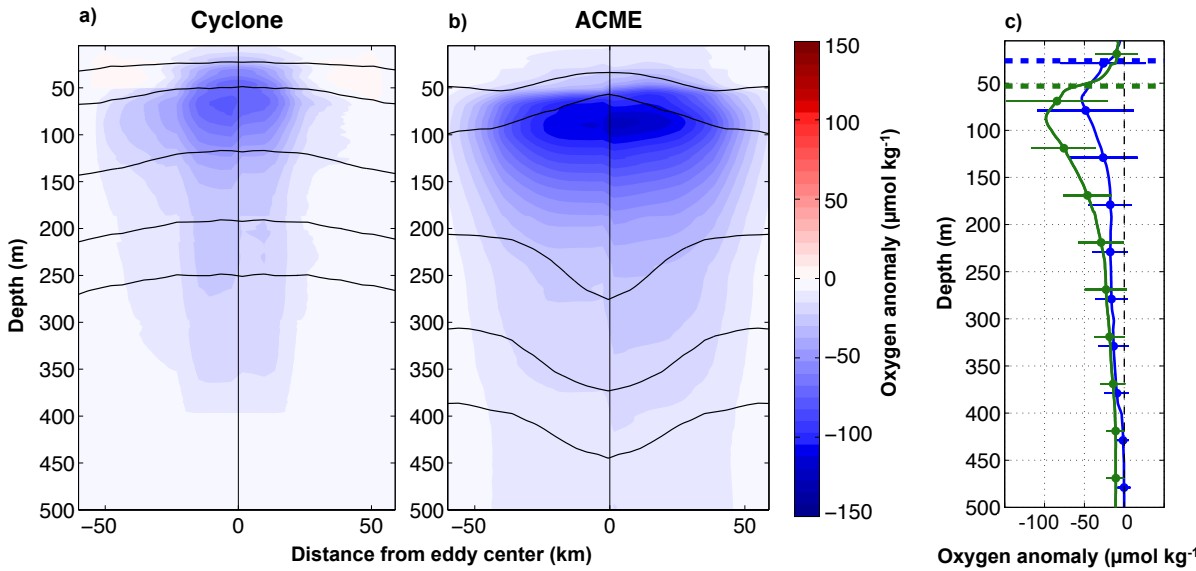

**Figure 8:** Vertical structure of oxygen from the composite **a)** CE and **b)** ACME in the ETNA presented as a half section across the eddies. The left side of both panels (-60 to 0 km) is based on reconstructed and measured oxygen profiles whereas the right side (0 to 60 km) is based on measured oxygen profiles only. Both methods are shown against the normalized radial distance. The grey lines represents the density surfaces inside the eddies. **c)** Mean profiles of the oxygen anomalies based on measured profiles only; green colors are associated to ACMEs and blue to CEs. Horizontal lines indicate the standard deviation of the oxygen anomaly at selected depths. The thick dashed lines indicates the mean ML within the different eddy types. The grey vertical dashed line represents zero oxygen.

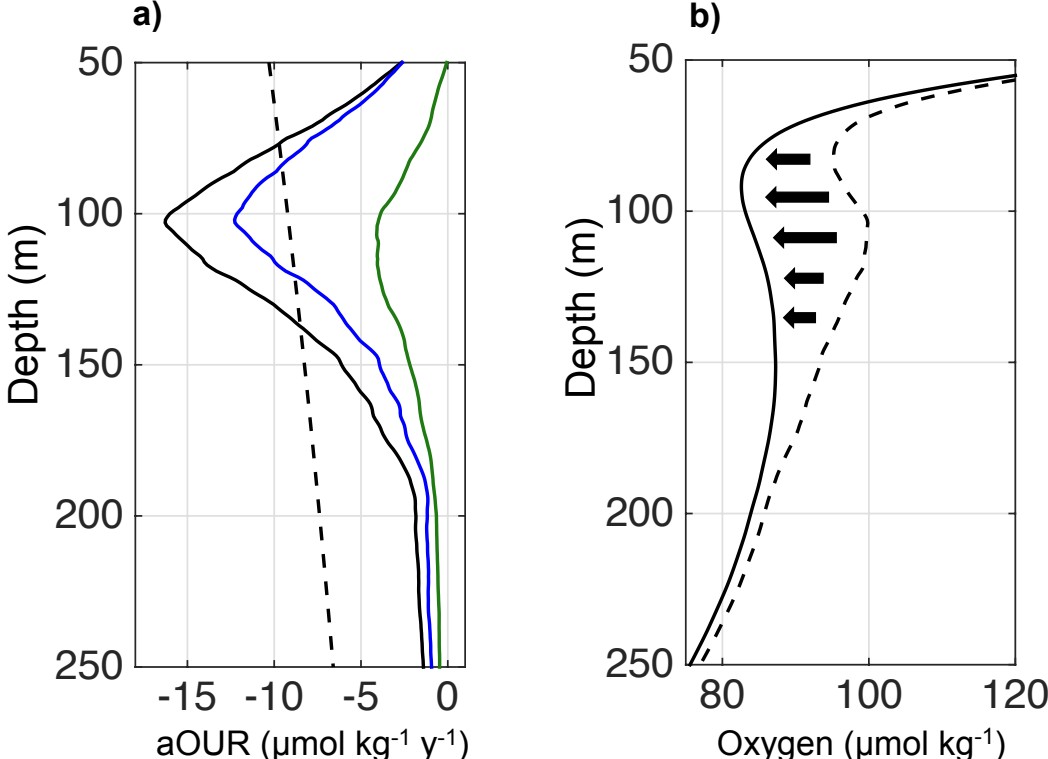

**Figure 9:** a) Depth profile of the apparent oxygen utilization rate (aOUR, µmol kg$^{-1}$ y$^{-1}$) for the Atlantic as published from Karstensen et al. (2008) (dashed black line). The oxygen consumption profile due to low-oxygen eddies referenced for the SOMZ region (solid black line) and the separation into CEs (blue) and ACMEs (green). The solid black line in b) represents the observed mean vertical oxygen profile of all profiles within the SOMZ against depth, whereas the dashed black line represents the theoretical vertical oxygen profile in the SOMZ without the dispersion of low-oxygen eddies. Naturally due to the dispersion of negative oxygen anomalies, the observed values (black line) are lower than the theoretical oxygen concentrations in the SOMZ without eddies (dashed black line). The impact of the dispersion of low-oxygen eddies on the oxygen budget in the depth of the shallow oxygen minimum zone are also indicated by the thick black arrows.