# Peer review of "Characterization of "dead-zone" eddies in the eastern tropical"

_Biogeosciences, 2016_

## Referee Comment (RC1) · Anonymous Referee #1 · 6 Apr 2016

**Summary**

Schütte et al. use an extensive compilation of observation based data comprising of shipboard measurements, mooring data, Argo float profiles, glider data as well as satellite based products to characterize mesoscale activity in the Eastern Tropical North Atlantic (ETNA). In particular, their analysis focuses on cyclonic eddies (CE) and anticyclonic modewater eddies (ACMEs), the associated oxygen depletion within these mesoscale structures and their potential contribution to the pronounced low oxygen environment within the shadow zone in the ETNA with the subtropical gyre to the North and the equatorial region to the South. They find that almost all observations of low oxygen concentrations below a canonical value of 40 $\mu$mol kg$^{-1}$ are co-located with either CEs or ACMEs that show negative oxygen anomalies which are most

pronounced right beneath the mixed layer. These anomalies are attributed both to high productivity in the surface waters and the subsequent respiration of organic material as well as to the dynamically induced isolation of the mesoscale structures with respect to lateral oxygen resupply. The authors conclude that the investigated eddies represent en essential part of the total consumption in the open ocean of the ETNA and partly contribute to the shallow low oxygen environment in the investigated region.

**1   General comments**

The presented work extends and complements previous work carried out by the community and the authors. In particular, the compilation of different observation based and quality-controlled data sources that extend previous records allow the authors to draw conclusions on the general characteristics and oxygen depletion within CEs and ACMEs in the studied region that advances our scientific understanding of mesoscale structures and their contribution to the mean distribution of biogeochemical properties. Moreover, the work is generally well-written, well-structured and results are presented in a clear and concise way. In my opinion, this manuscript thus represents work that is well suited for publication within the scope of Biogeosciences. Nevertheless, of course, I would like to make some comments and suggestions that should be addressed before publication and hopefully help the authors to further improve their work.

**A) The use of the term "dead zone"**
The authors use the term "dead zone" as a very prominent catchword throughout the whole manuscript. This term serves its purpose, but in my opinion, its use is not unproblematic. I think the use of this catchword is very colloquial and does not

acknowledge our scientific understanding of hypoxic environments that still provide habitats to specifically adapted species. Thus, it might potentially lead to premature interpretations and misunderstandings. To avoid these challenges, my suggestion is that the authors concentrate on phrasings such as "anoxic" and "hypoxic" and do not use "dead zone" in this context. If this term is used, it needs to be motivated, most importantly, but also discussed in the introduction in a more differentiated manner and the difficulties involved with interpreting such a catchphrase need to be appropriately addressed. In addition to specifically adapted species making use of these environments, marine organisms experience a highly non-linear sensitivity to low oxygen concentration and thresholds for hypoxia vary greatly among marine taxa (Keeling et al. 2010, Vaquer-Sunyer and Duarte 2008). A more elaborate motivation and differentiated discussion of the term can for example be found in the introduction of the review paper by Keeling et al. (2010) (see References at the end).

**B) Quantification, Significance, Relevance and Implications**
In my opinion, the presentation of some results in the current manuscript could be strengthened by clarifying certain paragraphs, putting results into a broader context and touch upon the relevance and potential implications of this work for other studies and concepts. Putting the results into a broader context can help a non-expert in mesoscale oxygen dynamics to better understand the relevance of this work. Reviewing some parts of the draft could add to the work presented here. Even though this is a major comment, let me get a little bit more specific here, to better convey my request:

Page 1, Line 24:
"increased consumption within these eddies represents an essential part of the total consumption...". First of all, I think that this specific sentence of the abstract could benefit from some quantification. Second, in the discussion (Page 11, Line 18) you

present the results from your budget analysis of the SOMZ oxygen consumption, stating that mesoscale structures contribute to about 6% of the observed low oxygen distribution. Even though this value is probably underestimating the total effect, as you argue in your work, 6% is not an essential part, in my opinion (please correct me if I misunderstood the line of argumentation). I think it's important that these paragraphs (abstract, discussion and conclusion) reflect each other and causal conclusions are drawn and described in a way that numbers and descriptions add up to the whole picture, even if this means being careful with catchwords such as "essential" or "significant". (Wouldn't a phrasing such as "the investigated contribution of mesoscale eddies only amounts to 6% of the observed low oxygen in the SOMZ. This value, though, is very likely to be underestimated due to..." also reflect the results but be more consistent when comparing the numerical and descriptive presentation?)

Page 8, Lines 20-21:
Can these estimates of oxygen consumption be put into the context of other observations, studies or estimates? How do these values in general compare with available estimates of average oxygen consumption? Are the results presented in the order of magnitude that the authors expected them to be, or is the effect stronger/weaker than what the authors expected? The way the results are presented here makes it hard for the reader to understand the magnitude of the mesoscale effect. Providing more context and comparisons would really help here.

Page 11, Lines 8-26:
This is a very important part of your work. I think it could be strengthened by re-phrasing some parts, putting the numbers into a broader context by providing comparisons that help the reader to better understand the magnitude of the discussed effects, and consistently present these findings in the abstract and conclusions (see comment above). I think this budget estimation is a central part of your work and very

well motivated on page 2 (lines 39-40), thus, in my opinion, it should be mentioned in the conclusions and the abstract. Please note the technical comments below to correct errors in this paragraph that, unfortunately, hinder the clear communication of these results.

Last but not least, your work naturally has implications for the nitrogen cycle. I am aware of some of the co-authors having submitted a manuscript on this issue as well (Karstensen et al. 2016). Nevertheless, I think it might help to at least mention some of the major implications for the nitrogen cycling within these mesoscale structures and the whole investigated region. Interested readers of this work might expect the authors to at least touch upon this or refer to the relevant literature.

**2  Specific comments**

**A) Chosen threshold of 40 $\mu$mol kg$^{-1}$**
Given a more differentiated discussion of the term "dead zone" (see comment above), can the authors elaborate on why they chose the specific threshold of 40 $\mu$mol kg$^{-1}$ and whether and how they would expect their results to change when choosing, e.g. a higher threshold (e.g. 60 $\mu$mol kg$^{-1}$ as mentioned in Keeling et al. 2010)? Would that significantly change the number of eddies considered as "low oxygen eddies" and thus increase the investigated sample or even strengthen the results?

**B) Physical contribution to the observed anomalies**
In the abstract, the authors state that the most pronounced oxygen anomalies are found right beneath the mixed layer and that this signal has been attributed to a combination of high productivity in the eddies' surface waters and the isolation of their

cores with respect to oxygen resupply. I do agree on this reasoning. However, I would like to mention an additional effect that has not been discussed in the manuscript and potentially plays a role here. The mere fact that the strongest anomalies are found at the base of the mixed layer hints at a pure physical contribution to the observed anomalies. Since density structures are shifted within the investigated eddies, this results in shifting the oxycline (i.e. shifting the isopycnals) and thus creating an oxygen anomaly that is of pure physical origin. If this is the case, can the author at least discuss the contribution of this mechanism on the observed concentrations, and if possible comment on the strength of this effect?

**C) Preconditioning through coastal environment**
The presented apparent oxygen utilization rates range from about 0.1 (CEs) to 0.2 (ACMEs) $\mu$mol kg$^{-1}$ d$^{-1}$. Even if the mesoscale structures are completely isolated and propagate offshore for, let's say, 2 months, this results in a oxygen decrease of only 12 $\mu$mol kg$^{-1}$ compared to its initial oxygen concentration. It seems thus very challenging for this mechanism alone to cause "dead zone" eddies. I think it is important to note somewhere that not only do enhanced productivity in the mesoscale structures and their physical isolation cause these very low oxygen eddies, but that there is a substantial contribution to the generation of these structures from the coastal environment, where most of them originate from. The above mentioned oxygen consumption alone would never be strong enough to result in a "dead zone" eddy, if it hadn't evolved from waters already low in oxygen along the upwelling region. I think this preconditioning is an important piece of the whole picture and should be briefly discussed somewhere.

**D) The use of the term "accuracy" (Page 4, Lines 13, 17, 20 and 25)**
The use of the term "accuracy" in the discussed context on page 4 confused me. To my knowledge, this term refers to the closeness of a measurement to a standard

or known value with "high accuracy" referring to "close measurements" and "low accuracy" describing rather poor measurement results. In general, one thus aims at high accuracies when observing natural phenomena and comparing to standard values. Here, the authors argue that the measurement methods have a rather high accuracy, but then state very low absolute values. Since the authors are describing measurement errors in the corresponding paragraph, I suggest they at least consider re-phrasing the sentences to ease the reader's understanding (e.g. using the term measurement error). I am glad to learn something about the correct use of the term "accuracy", in case I am wrong here.

**E) Discussion of other mesoscale features (anticyclonic eddies)**

On page 4 (line 30), the authors mention that their work also includes anticyclonic eddies. This eddy type is however not mentioned again. Even though I understand that the oxygen dynamics in eddies are strongly asymmetric between cyclonic and anticyclonic eddies, I wonder whether there is a compensating effect of anticyclonic eddies that stronger ventilate the water column. Could the authors elaborate on this, and maybe include a very brief comment on this in the manuscript?

**F) Figure 7 and Figure 9:**

As I understand, Figure 7 depicts mean profiles of apparent oxygen utilization of all eddies derived from the corresponding initial and actual oxygen profiles assuming a linear oxygen consumption (correct me if I am wrong). According to the corresponding figure caption of Figure 9, this figure shows the same property ($\mu$mol kg$^{-1}$ yr$^{-1}$ instead of $\mu$mol kg$^{-1}$ d$^{-1}$ in Fig. 7). This confused me because the magnitude shown in these two figures does not compare well. Can the authors comment on the difference between the two figures, if necessary elaborate on the corresponding text (Page 11, Lines 2-4) to better differentiate between the two results and maybe adjust the figure captions to help the reader understand their difference?

**3   Technical corrections and minor issues**

What follows is a list of minor technicalities and other issues I noticed while reviewing. I kindly ask the authors to correct typos and misspellings, reply to my questions and at least consider suggestions and comments on the (re-)phrasing of some sentences that might help to improve the reader's understanding.

Page 1, Lines 24-25: consumption of what?

Page 2, Line 28: consumption of what?

Page 3, Line 4: The use of "However" in this sentence is rather confusing since it doesn't contrast to what has been said before. Suggestion: "Due to the absence of other ventilation pathways in this zone, the influence of "dead-zone" eddies on the shallow oxygen minimum budget may be important and a closer examination worth the effort."

Page 3, Lines 10-11: As mentioned above, the mere fact that the density structure changes within these structures might add a purely physical contribution to the observed anomalies. Thus, it is not only due to biogeochemical processes that the anomalies are strongest at 100m depth, but rather due to a combination of both a purely physical displacement of the oxycline and biogeochemical processes in the water column above. This sentence should be re-phrased.

Page 3, Line 35: as THE last modification

Page 4, Line 27: as A final result

Page 4, Line 41: provided BY (phrasing of sentence is rather confusing)

Page 5, Line 7: data ARE considered (plural)

Page 5, Line 9: provided BY the NASA. The data WERE

Page 6, Line 1: Full stop missing (. . . propagation time is derived. We assume a mean. . .)

Page 6, Line 6: less saline and colder water than surrounding water

Page 6, Line 13: Depending on the status of isolation of the eddy, lateral mixing could take place (comma missing)

Page 7, Line 13: At its closest, the eddy center was . . . (comma missing)

Page 7, Line 18: blank space in unit missing

Page 7, Line 22: westward PROPAGATING eddy

Page 7, Line 37: data REVEAL (plural)

Page 8, Lines 26-27: If Figures 8 really depict normalized radial distances (as I assume), I suggest this is mentioned not only in the text, but also in the figure caption. Maybe the axis labeling needs to be adjusted as well. The same comment goes for Figure 6.

Page 9, Line 6: for THE ETNA

Page 9, Line 20: As discussed in Schütte et al. (2015), in case . . . (comma missing)

Page 10, Line 6: In the discussed context of eddy generation mechanisms, this formulation could be a little bit confusing, i.e. the word "generate" could be confused with eddy generation. Suggestion: I assume the authors would like to say "However, both eddy regimes feature eddies which locally ESTABLISH open ocean upwelling systems with high productivity at the surface and enhanced respiration beneath the ML during their westward propagation."

Page 11, Line 2: each year are propagate from the upwelling system near the coast
into the SOMZ and dissipate THERE.

Page 11, Line 8-10: This sentence should be re-phrased.

Page 11, Lines 16-19: Lines 16-19 (Attributing the oxygen concentrations. . .) are lacking in clarity and don't convey the intended message. Line 17 has an unnecessary parenthesis. Needs to be corrected and re-phrased.

Page 17, Line 7: Maybe a reference to Table 1 might be useful here for more information on M97.

Page 17, Line 9: around 80m depth (not plural)

Page 18, Line 3: Map of THE ETNA

Page 22, Line 4: b) CEs (use the introduced acronym)

Page 22, Line 5: when compared TO the SLA and SST

**References**

Keeling, RF, Kortzinger A, Gruber N. 2010. Ocean deoxygenation in a warming world. Annual Review of Marine Science. 2:199-229.

Vaquer-Sunyer R, Duarte CM. 2008. Thresholds of hypoxia for marine biodiversity. Proc. Natl. Acad. Sci. USA. 105:15452–57

Karstensen, J et al. 2016. Upwelling and isolation in oxygen-depleted anticyclonic modewater eddies and implications for nitrate cycling. Biogeosciences Discuss., doi:10.5194/bg-2016-34.

---

## Referee Comment (RC2) · Anonymous Referee #2 · 30 Apr 2016

Main Comments

Based on a set of data from different platforms, the authors analyze the impact of mesoscale eddies in the formation of the shallow oxygen minimum in the eastern tropical North Atlantic (which differs from the deepest minimum located below 400 m, that characterize the oxygen minimum zone of that region). Another central idea of the work is that the shallow oxygen minimum ($\sim$ 80 m depth) observed in some kind of eddies, is not due to the transport of waters with low oxygen carried by the eddies from the coastal regions, but is generated by the internal dynamics, particularly in cyclonic and sub-surface anticyclonic eddies (or anticlone modewater eddies). Within both types of eddies, the shallow isopycnal surfaces (located about 70-100 m depth) rise, favoring biological productivity near the surface (documented by positive chlorophyll anomalies estimated from satellite observations). The export of organic matter back into the subsurface would, thus, result in a relatively high rate of respiration leading to the formation of a shallow minimum of dissolved oxygen. Eddies effectively may "accumulate" this effect by transporting the water as they move.

I think the paper is an important contribution to the understanding of the dynamics of the biogeochemistry in the study region and highlights the effects of a special class of eddies (ACME), which is possibly relevant to other regions where the presence of sub-surface anticyclonic eddies is frequent. The work is fairly well structured and in general, the argument is consistent and can be followed easily. It seems that the authors have done a good job and in my opinion is an important contribution to understanding the hydrography and the biogeochemistry in that region, and it is also a contribution on the role of mesoscale eddies in the ocean. However, there are two issues that seem to me that should be discussed:

(1) Subsurface anticiclonic eddies may not have a proper manifestation in satellite altimetry. For example, contrasting Figure 5a for the cyclonic eddy and that for the ACME (Figure 5b), the latter has very small speed anomalies near the surface, and thus the sea level (and geostrophic velocity) anomalies should be small. This should be a relatively major problem if geostrophic velocities, based on altimetry, are used to identify, define the contours of these eddies and to position oxygen profiles.

2) The authors argue that the water remains fairly isolated within eddies. Although several studies (based on observation, numerical modeling and theoretical models) have shown that this phenomenon is correct, this is generally true for high latitude or subtropical eddies. Eddies ability to trap and transport water could be lower in the more linear equatorial region. This should be an issue to consider, at least for the southern part of the study area, located south of 12 ° N.

Another (positive) comment is that given the extensive data set used in the study, the authors present quantitative information and in some cases, allows them to estimate statistical errors based on the standard deviation. In general, dissolved oxygen data

is relatively scarce in large areas of the open ocean, this work is undoubtedly also a contribution in this regard.

Other minor comments

In the first paragraph of the introduction, the references to support some general sentences do not seem to me the most appropriate (for example, lines 6, 7 and 8). I do not mean that the argument is fallacious (magister dixit), but I think there are other studies that might have greater authority to support what is mentioned.

P4. L 1-6. Time lag for optode sensors is rather long given important differences between glider dives and climbs. How were the optode data from gliders corrected.

Page 4 lines 14-15 and 22-23. Aanderaa optodes were really calibrated (I mean to change the calibration constants) using CTD cast or the casts were used to estimate the accuracy of the optodes.

P7. L24 (and 16). Salinity in the core of ACME is mentioned as an important variable, why did you decided not to show it.

---

## Author Comment (AC1) · 17 Jun 2016

**1** Final Author Comments**

**2 "Characterization of "dead-zone" eddies in the tropical Northeast Atlantic Ocean"**

3 Florian Schuette, Johannes Karstensen, Gerd Krahmann, Helena Hauss, Björn Fiedler,

4 Peter Brandt, Martin Visbeck and Arne Körtzinger

5 fschuette@geomar.de

6

7 Dear Editor, dear Reviewer,

8 We would like to thank you for the positive evaluation of our manuscript. The constructive 9 criticism, the corrections and suggestions surely helped to improve the manuscript. In the 10 following we address the remarks of the Reviewer #2 in detail and how we intend to address 11 his/her concerns in the manuscript. The comments by the reviewer are shown in *italic* and our 12 responses in normal text.

13

**14 Anonymous Referee #2**

**15 Main Comments**

Based on a set of data from different platforms, the authors analyze the impact of mesoscale 16 17 eddies in the formation of the shallow oxygen minimum in the eastern tropical North Atlantic 18 (which differs from the deepest minimum located below 400 m, that characterize the oxygen 19 minimum zone of that region). Another central idea of the work is that the shallow oxygen 20 minimum ( $\sim 80$  m depth) observed in some kind of eddies, is not due to the transport of 21 waters with low oxygen carried by the eddies from the coastal regions, but is generated by the 22 internal dynamics, particularly in cyclonic and subsurface anticyclonic eddies (or anticyclone 23 modewater eddies). Within both types of eddies, the shallow isopycnal surfaces (located about 24 70-100 m depth) rise, favoring biological productivity near the surface (documented by 25 positive chlorophyll anomalies estimated from satellite observations). The export of organic 26 matter back into the subsurface would, thus, result in a relatively high rate of respiration 27 leading to the formation of a shallow minimum of dissolved oxygen. Eddies effectively may

1 "accumulate" this effect by transporting the water as they move.

2 I think the paper is an important contribution to the understanding of the dynamics of the 3 biogeochemistry in the study region and highlights the effects of a special class of eddies 4 (ACME), which is possibly relevant to other regions where the presence of subsurface 5 anticyclonic eddies is frequent. The work is fairly well structured and in general, the 6 argument is consistent and can be followed easily. It seems that the authors have done a good 7 job and in my opinion is an important contribution to understanding the hydrography and the 8 biogeochemistry in that region, and it is also a contribution on the role of mesoscale eddies in 9 the ocean. However, there are two issues that seem to me that should be discussed:

10 - Thank you very much for this positive evaluation.

(1) Subsurface anticiclonic eddies may not have a proper manifestation in satellite altimetry.
For example, contrasting Figure 5a for the cyclonic eddy and that for the ACME (Figure 5b),
the latter has very small speed anomalies near the surface, and thus the sea level (and
geostrophic velocity) anomalies should be small. This should be a relatively major problem if
geostrophic velocities, based on altimetry, are used to identify, define the contours of these
eddies and to position oxygen profiles.

17 - It is correct that ACMEs have a weak surface signature, which makes them more difficult to 18 be detected and tracked by satellite altimetry compared to normal anticyclonic/cyclonic 19 eddies. In the present analysis only eddies detected with a common Sea Level Anomaly 20 (SLA) threshold are followed with the tracking algorithms. Resulting eddy composites of 21 SLA, Sea Surface Temperature (SST) and Seas Surface Salinity (SSS) are shown in Figure 1. 22 The weaker anomaly of ACMEs compared to the other types of eddies is apparent. However, 23 as there should exist also ACMEs with weak or even no SLA signature, we expect that the 24 frequency of occurrence of ACMEs is underestimated. We included a corresponding 25 statement in the text.

26

---

## Author Response (AR1)

**"Characterization of "dead-zone" eddies in the eastern tropical North Atlantic"**

Florian Schuette, Johannes Karstensen, Gerd Krahmann, Helena Hauss, Björn Fiedler, Peter Brandt, Martin Visbeck and Arne Körtzinger fschuette@geomar.de

Dear Editor,

We would like to thank you for the overall positive evaluation of our manuscript and your remarks, which will surely help to improve the manuscript. We try to strengthen the manuscript by clarifying some paragraphs (for example putting results into a broader context) in particular in the introduction and discussion. In the following we address your remarks and how we intend to address the concerns in the manuscript. After that we include the Final Author comments of the reviewer #1 and reviewer #2. At the end a marked-up version of the manuscript is attached.

**Comments of Denis Gilbert**

**1.**

*P. 8, lines 20-21 of original manuscript: The confidence interval for the CE oxygen trend includes zero (0.10 ± 0.12). Given this, you cannot say that oxygen is decreasing within the CE. I propose that this sentence might be rephrased as "On average the oxygen concentration decreases by about $0.19 \pm 0.08$ $\mu mol$ $kg^{-1}$ $d^{-1}$ in the core of an isolated ACME, but has no significant trend in the core of an isolated CE ($0.10 \pm 0.12$ $\mu mol$ $kg^{-1}$ $d^{-1}$)".*

- Yes, that is true. We decided to take your suggested sentence and changed p: 9, lines 39-40 to "On average the oxygen concentration decreases by about $0.19 \pm 0.08$ $\mu mol$ $kg^{-1}$ $d^{-1}$ in the core of an isolated ACME, but has no significant trend in the core of an isolated CE ($0.10 \pm 0.12$ $\mu mol$ $kg^{-1}$ $d^{-1}$)".

**2.**

*Referee # 1's dislike of the term "dead-zone" is shared by many scientists because these low-oxygen waters are certainly not devoid of life. Given this, please make sure that double quotation marks always accompany the expression "dead-zone" in the final version of the manuscript, so that people understand this expression is a just a metaphor.*

- We decide to substitute nearly all "dead-zones" within the manuscript through low-oxygen eddies. If we used the word "dead-zone" we always accompany the expression with double quotation marks.

**3.**

*In your response to Referee # 1 (p.8, lines 24-26), you mention your rationale for picking a 40 $\mu mol$ $kg^{-1}$ $d^{-1}$*

*hypoxic threshold. Please include this rationale in the revised manuscript.*

- We included the explanation why we choose 40 µmol µmol kg$^{-1}$ as a threshold at two positions in the
manuscript. P. 2. lines 8-9:

"Traditionally the ETNA is considered to be "hypoxic", with minimal oxygen concentrations of marginally
below 40 µmol kg$^{-1}$ (e.g. Stramma et al. (2009)) (Fig. 1a)."

and P. 27 lines 28-33:

"The pelagic zones of the ETNA are traditionally considered to be "hypoxic", with minimal oxygen
concentrations of marginally below 40 µmol kg$^{-1}$ (Brandt et al., 2015; Karstensen et al., 2008; Stramma et al.,
2009). This is also true for the upper 200 m (Fig. 1). However, single oxygen profiles taken from various
observing platforms (ships, moorings, gliders, floats) with oxygen concentrations in the range of severe hypoxia
(< 20 µmol kg$^{-1}$) and even anoxia (~ 1 µmol kg$^{-1}$) conditions and consequently below the canonical value of 40
µmol kg$^{-1}$ (Stramma et al., 2008) are found in a surprisingly high number (in total 180 profiles) in the ETNA."

**4.**
*In your response to Referee # 1 (p.18, lines 10-11), I find this proposed new sentence confusing, especially this*
*bit: "analog to the SLA".*

- We rephrased the sentence, P.5 lines 17-18 to:

"The SLA and geostrophic velocity anomalies also provided by AVISO were chosen for the time period January
1998 to December 2014."

**5.**
*In Figure 2 of the authors' reply to anonymous referee #1, please add text labels to the black isopycnal contour*
*lines.*

- Done.

[Figure]

**Figure 2:** Oxygen in µmol kg$^{-1}$ (color) section along 18°N on the Mauritanian shelf conducted from the RS Meteor cruise M107 in June 2014. Black lines represent density, grey diamonds at the top of the figure locate the positions of the individual CTD casts.

**6.**

*The new Figure 1 included in your response to Referee # 2 presents useful additional information relative to Figure 6 of your original manuscript. You might like to consider producing a figure that would present SLA, SST, SSS and Chl a composites for cyclones, anticyclones and ACMEs, thus combining the information found in both of these figures.*

- We decided not to show a figure, which includes Sea Surface Salinity (SSS) and the surface signatures from "normal" anticyclones because of several reasons: First, we published such a figure in Schütte et al., 2016 "Occurrence and characteristics of mesoscale eddies in the tropical northeastern Atlantic" (Figure 11). Second, we did not mention "normal" anticyclones in the manuscript and third we did not introduce or use SSS in the manuscript.

**7.**

*On page 6 of your response to Referee # 2, you wrote that you changed figure 5 of the original manuscript by substituting the temperature with salinity. I suggest that in the revised manuscript, this particular figure should present both temperature and salinity panels in addition to meridional velocity and oxygen. Also, I must say that I preferred seeing the oxygen contours of the original figure 5, as they present more information than only oxygen concentration at a nominal depth of 120 m that you presented in Figure 2 or your response to Referee # 2.*

- We changed the figure and now show velocity, salinity, temperature and oxygen. But we decide to not show the oxygen contours, because it is based only on one instrument in 120 m depth. In the original figure I assumed saturation at the surface and interpolated in between, but my Co-Authors mentioned that it is not correct to show contourlines based on only one measuring device.

[Figure]

[Figure]

**Figure 5:** Meridional velocity, temperature, salinity and oxygen of an exemplary **a)** CE and **b)** ACME at the CVOO mooring. Both eddies passed the CVOO on a westward trajectory with the eddy center north of the mooring position (CE 20 km, ACME 13 km). The CE passed the CVOO from October to December 2006 and the ACME between January and March 2007. The thick black lines in the velocity plots indicate the position of an upward looking ADCP. Below that depth calculated geostrophic velocity is shown. The white lines represent density surfaces inside the eddies and the thin grey lines isolines of temperature and salinity, respectively. Thin black lines in the temperature and salinity plot mark the vertical position of the measuring devices. On the right time series of oxygen is shown from the one sensor available at nominal 120 m depth.

**Anonymous Referee #1**

*Summary*

*Schütte et al. use an extensive compilation of observation based data comprising of shipboard measurements, mooring data, Argo float profiles, glider data as well as satellite based products to characterize mesoscale activity in the Eastern Tropical North Atlantic (ETNA). In particular, their analysis focuses on cyclonic eddies (CE) and anticyclonic modewater eddies (ACMEs), the associated oxygen depletion within these mesoscale structures and their potential contribution to the pronounced low oxygen environment within the shadow zone in the ETNA with the subtropical gyre to the North and the equatorial region to the South. They find that almost all observations of low oxygen concentrations below a canonical value of 40 µmol/kg are co-located with either CEs or ACMEs that show negative oxygen anomalies which are most pronounced right beneath the mixed layer. These anomalies are attributed both to high productivity in the surface waters and the subsequent respiration of*

*organic material as well as to the dynamically induced isolation of the mesoscale structures with respect to lateral oxygen resupply. The authors conclude that the investigated eddies represent en essential part of the total consumption in the open ocean of the ETNA and partly contribute to the shallow low oxygen environment in the investigated region.*

*1 General comments*

*The presented work extends and complements previous work carried out by the community and the authors. In particular, the compilation of different observation based and quality-controlled data sources that extend previous records allow the authors to draw conclusions on the general characteristics and oxygen depletion within CEs and ACMEs in the studied region that advances our scientific understanding of mesoscale structures and their contribution to the mean distribution of biogeochemical properties. Moreover, the work is generally well-written, well-structured and results are presented in a clear and concise way. In my opinion, this manuscript thus represents work that is well suited for publication within the scope of Biogeosciences. Nevertheless, of course, I would like to make some comments and suggestions that should be addressed before publication and hopefully help the authors to further improve their work.*

- Thank you very much for this evaluation.

*A) The use of the term "dead zone"*

*The authors use the term "dead zone" as a very prominent catchword throughout the whole manuscript. This term serves its purpose, but in my opinion, its use is not unproblematic. I think the use of this catchword is very colloquial and does not acknowledge our scientific understanding of hypoxic environments that still provide habitats to specifically adapted species. Thus, it might potentially lead to premature interpretations and misunderstandings. To avoid these challenges, my suggestion is that the authors concentrate on phrasings such as "anoxic" and "hypoxic" and do not use "dead zone" in this context. If this term is used, it needs to be motivated, most importantly, but also discussed in the introduction in a more differentiated manner and the difficulties involved with interpreting such a catchphrase need to be appropriately addressed. In addition to specifically adapted species making use of these environments, marine organisms experience a highly non-linear sensitivity to low oxygen concentration and thresholds for hypoxia vary greatly among marine taxa (Keeling et al. 2010, Vaquer-Sunyer and Duarte 2008). A more elaborate motivation and differentiated discussion of the term can for example be found in the introduction of the review paper by Keeling et al. (2010) (see References at the end).*

- Thank you very much for pointing this out and reminding us to be more precise about the term "dead-zone" eddies. We totally agree with the reviewer that the use of the catchword "dead-zone" is problematic and imprecise. However, this term is chosen to be a major topic of the special issue and is consequently used in all of the associated manuscripts. We do not want to exclude us from that community and decided to use that term as well. In the understanding of the special issue a "dead-zone" is more a phenomenon than a certain concentration level and created by the variability in oxygen - in particular a "sudden" decrease ("sudden" with respect to life/adaption cycles of organisms). The "sudden" decrease in oxygen forces organisms to leave a region (if they are able to) or to die (the dead in "dead-zone"). This phenomenon is described for limnic and coastal systems and, as introduced in Karstensen et al. (2015), can occur in the open ocean in isolated eddies as well.

A more detailed discussion referring the used oxygen threshold in the manuscript and the mentioned paper by Keeling et al. (2010) is also given below (page 8 and line 24-30). However, we agree with the reviewer that a more differentiated introduction of the term "dead-zone" is certainly needed in our manuscript. We insert a paragraph in the introduction at page 2 line 28:

"The majority of organisms are insensitive to different oxygen levels as long as concentrations are high enough (Keeling et al. 2010). However, as soon as the oxygen falls below a certain critical threshold (which varies between different organisms) the most organisms suffer from a variety of stresses, which can lead to death if they are not able to migrate elsewhere and critical concentrations persists for too long (Gray et al. 2002, Keeling et al. 2010). It could be shown that the observed oxygen depleted eddy cores have profound impacts on microbial (Löscher et al. 2015) and metazoan (Hauss et al., 2016) communities. Furthermore the oxygen depleted cores of these eddies evolve in relatively "short" time scales ("short" with respect to time scales of life/adaption cycles of organisms), which resembles an environment similar to the "dead-zone" formation in coastal areas and lakes. Consequently, these oxygen depleted eddies have been termed "dead-zone" eddies (for a more detailed definition see also Karstensen et al., 2015)."

**B) Quantification, Significance, Relevance and Implications**

*In my opinion, the presentation of some results in the current manuscript could be strengthened by clarifying certain paragraphs, putting results into a broader context and touch upon the relevance and potential implications of this work for other studies and concepts. Putting the results into a broader context can help a non-expert in mesoscale oxygen dynamics to better understand the relevance of this work. Reviewing some parts of the draft could add to the work presented here.*

- Thank you very much for the assessment of our results. We worked through your following suggestions and tried in the complete manuscript to clarify some parts and to bring the results into a broader context to show the relevance and implication for other studies.

*Even though this is a major comment, let me get a little bit more specific here, to better convey my request:*

*Page 1, Line 24:*

*"increased consumption within these eddies represents an essential part of the total consumption. . .". First of all, I think that this specific sentence of the abstract could benefit from some quantification. Second, in the discussion (Page 11, Line 18) you present the results from your budget analysis of the SOMZ oxygen consumption, stating that mesoscale structures contribute to about 6% of the observed low oxygen distribution. Even though this value is probably underestimating the total effect, as you argue in your work, 6% is not an essential part, in my opinion (please correct me if I misunderstood the line of argumentation). I think it's important that these paragraphs (abstract, discussion and conclusion) reflect each other and causal conclusions are drawn and described in a way that numbers and descriptions add up to the whole picture, even if this means*

*being careful with catchwords such as "essential" or "significant". (Wouldn't a phrasing such as "the investigated contribution of mesoscale eddies only amounts to 6% of the observed low oxygen in the SOMZ. This value, though, is very likely to be underestimated due to..." also reflect the results but be more consistent when comparing the numerical and descriptive presentation?)*

- That is right. We totally agree that the 6% are misleading as they suggest only a small impact of "dead-zone" eddies on the oxygen concentration in the ETNA region. The 6% are related to the absolute oxygen concentration (125 $\mu$mol kg$^{-1}$). More interesting is the impact of "dead-zone" eddies on the existence of the shallow OMZ. Hence, the oxygen anomaly due to "dead-zone" eddies should be related to the strength of the shallow OMZ, whereas the latter is defined as the difference between the profile neglecting the shallow OMZ and the actual profile, which is observed. Relating these values results in dead zone eddies being responsible for around 25% of the shallow OMZ. Thus we have eliminated the value of 6% throughout the manuscript and replaced it with the absolute contribution of the "dead-zone" eddies, which is a reduction of 7 $\mu$mol kg$^{-1}$.

Furthermore we changed the abstract at the position mentioned by the reviewer at page 1 line 24 from:

"The locally increased consumption within these eddies represents an essential part of the total consumption in the open tropical Northeast Atlantic Ocean and might be partly responsible for the formation of the shallow oxygen minimum zone."

to

"The locally increased oxygen consumption within the eddy cores enhanced the total consumption in the in the open tropical Northeast Atlantic Ocean and might be partly responsible for the formation of the shallow oxygen minimum zone."

*Page 8, Lines 20-21:*

*"On average the oxygen concentration in the core of an isolated CE (ACME) decreases by about 0.10 (0.19) ± 0.12 (0.08) $\mu$mol kg$^{-1}$ d$^{-1}$."*

*Can these estimates of oxygen consumption be put into the context of other observations, studies or estimates? How do these values in general compare with available estimates of average oxygen consumption? Are the results presented in the order of magnitude that the authors expected them to be, or is the effect stronger/weaker than what the authors expected? The way the results are presented here makes it hard for the reader to understand the magnitude of the mesoscale effect. Providing more context and comparisons would really help here.*

- These numbers are classified and discussed in the section *4. Discussion* at page 10 line 8 to 14:

"In combination with the eddy dynamics and its associated isolation of the CE (ACME) core, the oxygen content is decreasing on average by about 0.10 (0.19) ± 0.12 (0.08) $\mu$mol kg$^{-1}$ d$^{-1}$ in the ETNA. The apparent oxygen utilization rate (aOUR) is based on 504 oxygen measurements in CEs and ACMEs. It is in the range of recently published aOUR estimates for CEs (Karstensen et al., 2015) and ACMEs (Fiedler et al., 2016) based on single measurements in "dead-zone" eddies. An important point regarding the method to derive the aOURs is the initial coastal oxygen concentration, which is highly variable in coastal upwelling regions (Thomsen et al., 2015). "

But we agree that these numbers are difficult to be classified by the reader in the first place. We expand the sentence, at page 8 line 22, to give the reader a first idea about the magnitude of the aOUR estimates:

"On average the oxygen concentration in the core of an isolated CE (ACME) decreases by about 0.10 (0.19) $\pm$ 0.12 (0.08) $\mu$mol kg$^{-1}$ d$^{-1}$ which is in the range of recently published aOUR estimates for CEs (Karstensen et al., 2015) and ACMEs (Fiedler et al., 2016)."

*Page 11, Lines 8-26: This is a very important part of your work. I think it could be strengthened by rephrasing some parts, putting the numbers into a broader context by providing comparisons that help the reader to better understand the magnitude of the discussed effects, and consistently present these findings in the abstract and conclusions (see comment above).*

- We rephrased the mentioned paragraph, improved the structure and hopefully clarifying the description of the used budget estimation:

"Instead of describing the effect of the dead-zone eddies on the oxygen consumption we now consider a box model approach for the SOMZ. The basis of this box model is the mixing of higher oxygen waters (the background conditions) with lower oxygen waters (the "dead-zone" eddies). The average oxygen concentrations within the eddies in the considered depth range, i.e. 50 to 150 m depth, are 73 (66) $\mu$mol kg$^{-1}$ for CEs (ACMEs). The average oxygen concentration of the background field averaged over the same depth range (between 50 and 150 m depth) derived from the MIMOC climatology (Schmidtko et al. (2013)) is 118 $\mu$mol kg$^{-1}$. This climatological value includes the contribution of low oxygen eddies. If we now consider the respective oxygen concentrations and volumes of the SOMZ and the eddies (multiplied by their frequency of occurrence per year), we are able to calculate the theoretical background oxygen concentration for the SOMZ without eddies to be 125 $\mu$mol kg$^{-1}$. Naturally due to the dispersion of negative oxygen anomalies, the oxygen concentrations in the SOMZ without eddies must be higher than the observed climatological values. Attributing the difference of these oxygen concentrations on the one hand in the SOMZ without eddies (125 $\mu$mol kg$^{-1}$) and on the other hand the observed climatological values in the SOMZ with eddies (118 $\mu$mol kg$^{-1}$), solely to the decrease induced by the dispersion of eddies, we find that a reduction of around 7 $\mu$mol kg$^{-1}$ of the observed climatological oxygen concentration in the SOMZ box can be associated with the dispersion of eddies. Consequently, the oxygen consumption in this region is a mixture of the large-scale metabolism in the open ocean (Karstensen et al. 2008) and the enhanced metabolism in low oxygen eddies (Karstensen et al. 2016, Fiedler et al. 2015)."

*I think this budget estimation is a central part of your work and very well motivated on page 2 (lines 39-40), thus, in my opinion, it should be mentioned in the conclusions and the abstract. Please note the technical comments below to correct errors in this paragraph that, unfortunately, hinder the clear communication of these results.*

- That is right, we mention now the results in the abstract on page 1 and line 25:

"In a simple box model approach the investigated contribution to the observed low oxygen in the shallow oxygen minimum zone of "dead-zone" eddies is a reduction of the oxygen concentration of 7 µmol kg$^{-1}$.""

And in the conclusion on page 12 and line 3:

"A simple box model approach on the basis of mixing ratios of high oxygen waters with low oxygen waters in the SOMZ reveals that a reduction of 7 µmol kg$^{-1}$ of the observed oxygen in the shallow oxygen minimum zone is explainable due to the existence of "dead-zone" eddies. This value, though, is very likely to be underestimated due to difficulties in identifying and tracking of ACMEs."

*Last but not least, your work naturally has implications for the nitrogen cycle. I am aware of some of the*

*coauthors having submitted a manuscript on this issue as well (Karstensen et al. 2016). Nevertheless, I think it*

*might help to at least mention some of the major implications for the nitrogen cycling within these mesoscale*

*structures and the whole investigated region. Interested readers of this work might expect the authors to at least*

*touch upon this or refer to the relevant literature.*

- That is correct, we insert a paragraph to give in the introduction some more details on the nitrogen cycle at page 2 line 28:

"The intense OMZ has profound impacts on microbial (Löscher et al. 2015) and metazoan (Hauss et al., 2016)

communities. While denitrification is usually absent from the open tropical Atlantic, the detection of nirS gene transcripts (the key functional marker for denitrification) in an ACME potentially indicated nitrogen loss processes in the oxygen depleted eddy core (Löscher et al., 2015). However, the close-to-Redfield N:P

stoichiometry in the same ACME (Fiedler et al., 2015), does not suggest a large-scale net loss of bioavailable nitrogen. In general, the relative magnitude of nutrient upwelling/primary productivity, nitrogen fixation and denitrification may vary between different eddies because of differences in the initial water mass in the eddies'

core, the eddies' age and the external forcing (in particular wind stress and dust/iron input)."

***2 Specific comments***

***A) Chosen threshold of 40 µmol/kg***

*Given a more differentiated discussion of the term "dead zone" (see comment above), can the authors elaborate*

*on why they chose the specific threshold of 40 µmol/kg and whether and how they would expect their results to*

*change when choosing, e.g. a higher threshold (e.g. 60 µmol/kg as mentioned in Keeling et al. 2010)? Would*

*that significantly change the number of eddies considered as "low oxygen eddies" and thus increase the*

*investigated sample or even strengthen the results?*

- We wanted to highlight profiles with anomalous low oxygen concentrations. The minimal dissolved oxygen concentrations in the ETNA are in the range of 40-50 µmol kg$^{-1}$, thus the 40 µmol kg$^{-1}$ threshold is chosen to clearly identify anomalous low oxygen concentrations. The number of profiles, probably near the coast or in the center of the OMZ, would increase if 60 µmol kg$^{-1}$ were chosen as threshold. But the majority of the profiles would not be associated to mesoscale eddies, as the oxygen values (40-60 µmol kg$^{-1}$) are appearing in the large-
scale oxygen distribution of the ETNA.

***B) Physical contribution to the observed anomalies***

*In the abstract, the authors state that the most pronounced oxygen anomalies are found right beneath the mixed*
*layer and that this signal has been attributed to a combination of high productivity in the eddies' surface waters*
*and the isolation of their cores with respect to oxygen resupply. I do agree on this reasoning. However, I would*
*like to mention an additional effect that has not been discussed in the manuscript and potentially plays a role*
*here. The mere fact that the strongest anomalies are found at the base of the mixed layer hints at a pure physical*
*contribution to the observed anomalies. Since density structures are shifted within the investigated eddies, this*
*results in shifting the oxycline (i.e. shifting the isopycnals) and thus creating an oxygen anomaly that is of pure*
*physical origin. If this is the case, can the author at least discuss the contribution of this mechanism on the*
*observed concentrations, and if possible comment on the strength of this effect?*

- That is a correct, a vertical displacement of isopycnals move lower oxygen concentrations closer to the mixed
layer. First we rephrased the sentence in the introduction at page 3 line 10-11 from:

"At about 100 m depth, biogeochemical processes further increase the nutrient and oxygen anomalies with
respect to the surrounding waters."

to

"At about 100 m depth, the elevated isopycnals in the eddies are associated to a displacement of the oxycline,
which brings lower oxygen concentrations closer to the mixed layer. Here biogeochemical processes further
increase the nutrient and oxygen anomalies with respect to the surrounding waters."

Further we investigated the contribution of the "physical" and "biogeochemical" part of the oxygen anomaly by
comparing the oxygen anomaly derived on density surfaces against the oxygen anomaly derived on isobars
(Figure 1).

[Figure]

**Figure 1:** Mean Oxygen anomaly of ACMEs (green) and CEs (blue) derived on isopycnal surfaces (dashed lines) and isobars (continuous lines). The anomaly on isopycnal surfaces (dashed lines) are derived, by building an oxygen anomaly of each eddy type on density surfaces. Afterwards a transformation in pressure coordinates is done referenced to a mean density profile from outside the eddy.

Derived on isobars the oxygen anomaly in the upper eddy core is more pronounced compared to the anomaly derived on isopycnals, due to the upward bending of the density surfaces. However, the maximal absolute values of the anomaly are nearly the same. Therefore we conclude that the pure "physical" effect of shifting the oxycline is much smaller than the "biogeochemical" part in crating the oxygen anomaly.

*C) Preconditioning through coastal environment*

*The presented apparent oxygen utilization rates range from about 0.1 (CEs) to 0.2 (ACMEs) µmol/kg/d. Even if the mesoscale structures are completely isolated and propagate offshore for, let's say, 2 months, this results in an oxygen decrease of only 12 µmol/kg compared to its initial oxygen concentration. It seems thus very challenging for this mechanism alone to cause "dead zone" eddies. I think it is important to note somewhere that not only do enhanced productivity in the mesoscale structures and their physical isolation cause these very low oxygen eddies, but that there is a substantial contribution to the generation of these structures from the coastal environment, where most of them originate from. The above mentioned oxygen consumption alone would never be strong enough to result in a "dead zone" eddy, if it hadn't evolved from waters already low in oxygen along the upwelling region. I think this preconditioning is an important piece of the whole picture and should be briefly discussed somewhere.*

- Yes that is right, the preconditioning due to low oxygen values at the shelf of the formation region was poorly described in the manuscript before. The reviewer mentioned correct that the preconditioning is an important part in the developing of the open ocean "dead-zone" eddies. We plotted the Shipboard CTD section with the lowest oxygen at the shelf of Mauretania and Senegal we could find (figure 2).

[Figure]

**Figure 2:** Oxygen in µmol kg$^{-1}$ (color) section along 18°N on the Mauritanian shelf conducted from the RS Meteor cruise M107 in June 2014. Black lines represent density, grey diamonds at the top of the figure locate the positions of the individual CTD casts.

An oxygen minimum is found directly in the core depth of the "dead-zone" eddies between 50 to 150 m with a locally occurrence of minimal oxygen concentrations of around 30-35 µmol kg$^{-1}$ very near to the shelf. Following the theory of the formation processes of ACMEs from McWilliams (1985) and D'Assaro (1988), these near-bottom shelf waters are most likely captured in the eddy cores. The isolated oxygen depleted eddy cores are thus a combination of already low oxygen concentrations from the beginning and the enhanced respiration associated to an oxygen loss with time. We added a paragraph at page 11 line 14 to discuss that in more detail:

"Regions with low oxygen concentrations around 30 µmol kg$^{-1}$ in the depth range between 50-150 m could locally identified at the shelf off Northwest Africa. However, all observed CEs or ACMEs contain a negative oxygen anomaly, partly because they transport water with initial low oxygen concentrations from the coast into the open ocean and additionally because the oxygen consumption in the eddies is more intense then in the surrounding waters (Karstensen et al. 2015a, Fiedler et al. 2015)."

***D) The use of the term "accuracy" (Page 4, Lines 13, 17, 20 and 25)***

*The use of the term "accuracy" in the discussed context on page 4 confused me. To my knowledge, this term*
*refers to the closeness of a measurement to a standard or known value with "high accuracy" referring to "close*

*measurements" and "low accuracy" describing rather poor measurement results. In general, one thus aims at high accuracies when observing natural phenomena and comparing to standard values. Here, the authors argue that the measurement methods have a rather high accuracy, but then state very low absolute values. Since the authors are describing measurement errors in the corresponding paragraph, I suggest they at least consider re-phrasing the sentences to ease the reader's understanding (e.g. using the term measurement error). I am glad to learn something about the correct use of the term "accuracy", in case I am wrong here.*

- That is right. Accuracy refers the closeness of a measurement to a known reference value. In our case we do not know the exact reference value, thus the usage of the word accuracy is not correct in that context. We used, as suggested from the reviewer, the word "measurement error" instead. Changes are made on page 4, lines 13: 17, 20 and 25:

lines 13: "The resulting measurement error were $\leq 1.5$ µmol $kg^{-1}$."

lines 17:     "We estimate their measurement error at $<3$ µmol $kg^{-1}$."

lines 20:     "The different manufacturers of Argo float oxygen sensors specify their measurement error at least better than 8 µmol $kg^{-1}$ or 5%, whichever is larger."

lines 25:     "We thus estimate their measurement error to about 3 µmol $kg^{-1}$."

*E) Discussion of other mesoscale features (anticyclonic eddies)*

*On page 4 (line 30), the authors mention that their work also includes anticyclonic eddies. This eddy type is however not mentioned again. Even though I understand that the oxygen dynamics in eddies are strongly asymmetric between cyclonic and anticyclonic eddies, I wonder whether there is a compensating effect of anticyclonic eddies that stronger ventilate the water column. Could the authors elaborate on this, and maybe include a very brief comment on this in the manuscript?*

- In this paper our main focus was to highlight sporadic profiles with very low oxygen concentrations between 50 to 150 m depth in the eastern tropical north Atlantic and that we could associate the profiles to CEs and ACMEs. We further tried to asses the number of such oxygen depleted eddies and the influence on the environment. Anticyclones play a minor role in the story. Furthermore, we think that the compensating effect of anticyclones is relatively small. The depression of isopycnals within anticyclones produces positive oxygen anomalies on depth levels, but on density surfaces these anomalies do not exist. To produce a compensating effect of anticyclones additional diapycnal processes are needed. Nevertheless we agree with the reviewer that during the decay of the eddy probably diapycnal processes are possible and therefore a compensation effect of anticyclones is not unlikely and should be mentioned and discussed in the paper.

First of all we delete the word anticyclone at page 4 line 30 as it is apparently confusing and unnecessary:

"To determine the characteristics of different eddy types from the assembled profiles, we separated them into

CEs, ACMEs and the "surrounding area" not associated with eddy-like structures following the approach of

Schütte et al. (2015). "

We further decided to discuss the influence on the oxygen budget of anticyclones on page 10 line 27:

"Anticyclonic rotating eddies with a low oxygen core are only observed for modewater type anticyclones (i.e.

ACMEs), but not for "normal" anticyclonic eddies which do not show an oxygen depleted eddy core. Instead, the downward bending of isopycnals within "normal" anticyclones produces positive oxygen anomalies on depth levels, whereas on density surfaces these anomalies do not exist."

and on page 12 line 26:

"In the contrary with additional diapycnal processes (for example during the decay of the eddy) a small compensating effect due to Anticyclones is expectable."

***F) Figure 7 and Figure 9:***

*As I understand, Figure 7 depicts mean profiles of apparent oxygen utilization of all eddies derived from the*

*corresponding initial and actual oxygen profiles assuming a linear oxygen consumption (correct me if I am*

*wrong). According to the corresponding figure caption of Figure 9, this figure shows the same property*

*(μmol/kg/yr instead of μmol/kg/d in Fig7). This confused me because the magnitude shown in these two figures*

*does not compare well. Can the authors comment on the difference between the two figures, if necessary*

*elaborate on the corresponding text (Page 11, Lines 2-4) to better differentiate between the two results and*

*maybe adjust the figure captions to help the reader understand their difference?*

- Thank you very much for this comment. We agree with the Reviewer that these pictures were confusing.

Hopefully we could clarify some parts with the following explanation and changes in the figure captions. Figure

3 showing both of the mentioned pictures of the Reviewer.

Figure 3a shows the profiles of the apparent oxygen utilization rate of ACMEs and CEs per day in the ETNA

region. It is calculated, as mentioned right by the reviewer, by using the propagation time of each eddy and an initial coastal oxygen profile and assuming a linear oxygen consumption (based on depth layers). It gives an indication of how much the oxygen concentration in isolated ACMEs and CEs cores in the ETNA region is reduced due to enhanced respiration.

Whereas figure 3b shows a budget term, namely the oxygen loss profile due to "dead-zone" eddies in the subarea

"SOMZ" induced by the ACMEs and CEs on each ispopycnal (converted back to depth). The profiles are derived, by building an oxygen anomaly of each eddy type on density surfaces ($O_2'$). The derived anomalies are multiplied by the mean number of eddies dissipating in the SOMZ per year ($n$) and weighted by the area of the eddy compared to the total area of the SOMZ ($A_{SOMZ}$ = triangle in Fig. 1a of the manuscript). Differences in the mean isopycnal layer thickness of each eddy type and the SOMZ are considered by multiplying the result with the ratio of the mean Brunt-Väisälä frequency ($N^2$) outside and inside the eddy, resulting in an apparent oxygen utilization rate per year ($\mu mol\ kg^{-1}\ y^{-1}$) due to "dead-zone" eddies in the SOMZ on density layers:

$$aOUR = nO_2' \frac{\pi r_{Eddy}^2 N_{SOMZ}^2}{A_{SOMZ} N_{Eddy}^2}$$

where $r_{Eddy}$ is the mean radius of the eddies.

[Figure]

**Figure 3: a)** Depth profiles of a mean apparent oxygen utilization rate (aOUR, μmol kg[-1] d[-1]) within CEs (blue)

and ACMEs (green) in the ETNA region with associated standard deviation (horizontal lines). **b)** Depth profile of apparent oxygen utilization rate (aOUR, μmol kg[-1] y[-1]) for the Atlantic as published from Karstensen et al.

(2008) (dashed black line), the oxygen consumption profile due to "dead-zone" eddies in the SOMZ (solid black line) and the separation into CEs (blue) and ACMEs (green).

We changed the figure caption of figure 7 to:

**"**Depth profiles of a mean apparent oxygen utilization rate (aOUR, μmol kg[-1] d[-1]) within CEs (blue) and ACMEs (green) in the ETNA region with associated standard deviation (horizontal lines). Derived by using the propagation time of each eddy, an initial coastal oxygen profile and the assumption of linear oxygen consumption (based on depth layers)."

Furthermore we changed the figure caption of figure 9 to:

"Depth profile of the apparent oxygen utilization rate (aOUR, μmol kg[-1] y[-1]) for the Atlantic as published from

Karstensen et al. (2008) (dashed black line). The oxygen consumption profile due to "dead-zone" eddies referenced for the SOMZ (solid black line) and the separation into CEs (blue) and ACMEs (green)."

*3 Technical corrections and minor issues*

*What follows is a list of minor technicalities and other issues I noticed while reviewing. I kindly ask the authors to correct typos and misspellings, reply to my questions and at least consider suggestions and comments on the (re-)phrasing of some sentences that might help to improve the reader's understanding.*

*Page 1, Lines 24-25: consumption of what?*

- We changed page 1 line 24-25 from:

"The locally increased consumption within these eddies represents an essential part of the total consumption in the open tropical Northeast Atlantic Ocean and might be partly responsible for the formation of the shallow oxygen minimum zone."

to

 "The locally increased oxygen consumption within these eddies represents a part of the total oxygen consumption in the open tropical Northeast Atlantic Ocean and might be partly responsible for the formation of the shallow oxygen minimum zone."

*Page 2, Line 28: consumption of what?*

- We changed page 2 line 28 from:

"The ventilation and consumption processes of thermocline waters in the ETNA result in two separate oxygen minima (Fig. 1b): a shallow one with a core depth of about 80 m and a deep one at a core depth of about 450 m."

to

"The ventilation and oxygen consumption processes of thermocline waters in the ETNA result in two separate oxygen minima (Fig. 1b): a shallow one with a core depth of about 80 m and a deep one at a core depth of about 450 m."

*Page 3, Line 4: The use of "However" in this sentence is rather confusing since it doesn't contrast to what has been said before. Suggestion: "Due to the absence of other ventilation pathways in this zone, the influence of "dead-zone" eddies on the shallow oxygen minimum budget may be important and a closer examination worth the effort."*

- We changed page 2 line 28 from:

"However, due to the absence of other ventilation pathways, the influence of "dead-zone" eddies on the shallow oxygen minimum budget may be elevated and a closer examination worth the effort."

to

"Due to the absence of other ventilation pathways in this zone, the influence of "dead-zone" eddies on the shallow oxygen minimum budget may be important and a closer examination worth the effort."

*Page 3, Lines 10-11: As mentioned above, the mere fact that the density structure changes within these structures might add a purely physical contribution to the observed anomalies. Thus, it is not only due to biogeochemical processes that the anomalies are strongest at 100m depth, but rather due to a combination of both a purely physical displacement of the oxycline and biogeochemical processes in the water column above. This sentence should be re-phrased.*

- We rephrased the sentence page 3 line 10-11 from:

"At about 100 m depth, biogeochemical processes further increase the nutrient and oxygen anomalies with respect to the surrounding waters."

to

"At about 100 m depth, the elevated isopycnals in the eddies are associated to a displacement of the oxycline. In combination with the biogeochemical processes they further increase the nutrient and oxygen anomalies with respect to the surrounding waters."

*Page 3, Line 35: as THE last modification*

- done

*Page 4, Line 27: as A final result*

- done

*Page 4, Line 41: provided BY (phrasing of sentence is rather confusing)*

- We rephrase the sentence Page 4, Line 41 from:

"Data of the SLA and of the geostrophic velocities, derived from the SLA and also provided from AVISO, for the period January 1998 to December 2014 were chosen."

to

"Geostrophic velocities anomalies also provided by AVISO were chosen analog to the SLA for the period January 1998 to December 2014."

*Page 5, Line 7: data ARE considered (plural)*

- done

*Page 5, Line 9: provided BY the NASA. The data WERE*

- done

*Page 6, Line 1: Full stop missing (... propagation time is derived. We assume a mean. . .)*

- done

*Page 6, Line 6: less saline and colder water than surrounding water*

- done

*Page 6, Line 13: Depending on the status of isolation of the eddy, lateral mixing could take place (comma*
*missing)*

- done

*Page 7, Line 13: At its closest, the eddy center was . . . (comma missing)*

- done

*Page 7, Line 18: blank space in unit missing*

- done

*Page 7, Line 22: westward PROPAGATING eddy*

- done

*Page 7, Line 37: data REVEAL (plural)*

- done

*Page 8, Lines 26-27: If Figures 8 really depict normalized radial distances (as I assume), I suggest this is*
*mentioned not only in the text, but also in the figure caption. Maybe the axis labeling needs to be adjusted as*
*well.*

- That is correct, we add a sentence in the caption of figure 8 page 24, line 4-5:

"Oxygen anomalies derived by both methods are shown against the normalized radial distance."

*The same comment goes for Figure 6.*

In figure 6 we decided to use unscaled coordinates, because the majority of the selected low oxygen eddies was
of similar size.

*Page 9, Line 6: for THE ETNA*

- done

*Page 9, Line 20: As discussed in Schütte et al. (2015), in case . . . (comma missing)*

- done

*Page 10, Line 6: In the discussed context of eddy generation mechanisms, this formulation could be a little bit*
*confusing, i.e. the word "generate" could be confused with eddy generation. Suggestion: I assume the authors*
*would like to say "However, both eddy regimes feature eddies which locally ESTABLISH open ocean upwelling*
*systems with high productivity at the surface and enhanced respiration beneath the ML during their westward*
*propagation."*
- We rephrased the sentence page 3 line 10-11 from:
"However, both eddy regimes feature eddies which generate during their westward propagation locally open
ocean upwelling systems with high productivity at the surface and enhanced respiration beneath the ML."
to
"However, both eddy regimes feature eddies which locally establish open ocean upwelling systems with high
productivity at the surface and enhanced respiration beneath the ML during their westward propagation."
*Page 11, Line 2: each year are propagate from the upwelling system near the coast into the SOMZ and dissipate*
*THERE.*
- done

*Page 11, Line 8-10: This sentence should be re-phrased.*

- We rephrased these two sentences from:

"An equivalent view is, by investigating a simple mix ratio of higher with lower oxygen waters in a box model
approach of the SOMZ. When averaging the oxygen concentrations of the eddies in the considered depth range,
i.e. 50 to 150 m depth, a mean oxygen concentration of 73 (66) $\mu$mol kg$^{-1}$ for CEs (ACMEs) is derived."

to

"Instead of describing the effect of the dead-zone eddies on the apparent oxygen conditions as an enhancement
of the oxygen utilization as above is to consider a box model approach for the SOMZ. The basis of this box
model approach is simply considering the mixing ratio of higher oxygen waters (the ambient conditions) with
lower oxygen waters (the "dead-zone" eddies). The average oxygen concentrations within the eddies in the
considered depth range, i.e. 50 to 150 m depth, are 73 (66) $\mu$mol kg$^{-1}$ for CEs (ACMEs)."

*Page 11, Lines 16-19: Lines 16-19 (Attributing the oxygen concentrations. . .) are lacking in clarity and don't*

*convey the intended message. Line 17 has an unnecessary parenthesis. Needs to be corrected and re-phrased.*

- We rephrased these two sentences from:

"Attributing the  difference of these values (oxygen concentration respiration without eddies (125 μmol kg$^{-1}$) and observed values  with eddies (118 μmol kg$^{-1}$) solely decreased due to the dispersion of eddies, we find that around 6% of the observed oxygen concentrations in our box model can be associated to the dispersion of eddies."

to

"Attributing the difference of these oxygen concentrations on the one hand the SOMZ without eddies (125 μmol kg$^{-1}$) and on the other hand the observed climatological values in the SOMZ with eddies (118 μmol kg$^{-1}$), solely to the decrease induced by the dispersion of eddies, we find that around 6% of the observed climatological oxygen concentration in the SOMZ box can be associated with the dispersion of eddies."

*Page 17, Line 7: Maybe a reference to Table 1 might be useful here for more information on M97.*

- We repeated the information regarding M97 from table 1 in the figure caption of figure 1, page 17, line 7:

"The black crosses in **a)** indicate the position of the CTD stations taken during the research cruise M97 in boreal summer 2013, which are used to calculate the mean vertical oxygen profile shown in **b)**."

*Page 17, Line 9: around 80m depth (not plural)*

- done

*Page 18, Line 3: Map of THE ETNA*

- done

*Page 22, Line 4: b) CEs (use the introduced acronym)*

- done

*Page 22, Line 5: when compared TO the SLA and SST*

- done

[Figure]

**Figure 1:** Sea Level Anomaly (SLA), Sea Surface Temperature (SST) and Sea Surface Salinity anomalies of the
composite cyclone, anticyclone and ACME in the tropical Atlantic off northwest Africa. SLA (color) and the
associated geostrophic velocity (white arrows) are shown for each eddy type in **a)**, **b)** and **c)**; SST anomaly in **d)**,
**e)** and **f)**; and SSS anomaly in **g)**, **h)** and **i)**, respectively. The circles mark the mean eddy radius. Taken from
Schütte et al. (2016).

We now added a sentence on page 10 line 23 that point out the weakness in the statistic assessment:

"As discussed in Schütte et al. (2016) we expect that the number of ACMEs is underestimated because of the
possible existence of ACMEs with a weak surface signature in SLA data."

*(2) The authors argue that the water remains fairly isolated within eddies. Although several studies (based on*
*observation, numerical modeling and theoretical models) have shown that this phenomenon is correct, this is*
*generally true for high latitude or subtropical eddies. Eddies ability to trap and transport water could be lower*
*in the more linear equatorial region. This should be an issue to consider, at least for the southern part of the*
*study area, located south of 12 ° N.*

- Thank you for the comment, this is a very interesting point. In general we where surprised to detect long lived low oxygen eddies in the region south of 12°N. At this stage we simply have to accept the fact that the low oxygen levels are present in these eddies and, as we see from the T/S characteristics, the water seems not to originate from the eastern boundary region as it is the case for eddies found further north. Following the trajectories it seems that the ACMEs are generated in the open ocean somewhere in the region between 5°N and 7°N. However, the eddies seem to be isolated long enough (and respiration is intense enough) to generate an oxygen depleted core during their westward propagation. Clearly, further studies on their generation mechanism and their characteristics are required.

We added one sentence to discuss less isolation in lower latitudes at page 10 line 17-19:

"The occurrence of oxygen depleted eddies south of 12°N is rather astonishing, as due to the smaller Coriolis parameter closer to the equator the southern eddies should be more short-lived and less isolated compared to eddies further north."

*Another (positive) comment is that given the extensive data set used in the study, the authors present quantitative information and in some cases, allows them to estimate statistical errors based on the standard deviation. In general, dissolved oxygen data is relatively scarce in large areas of the open ocean, this work is undoubtedly also a contribution in this regard.*

- As mentioned right by the reviewer the dissolved oxygen data is relatively scare and flawed with large errors (Argo-floats) in wide areas of the open ocean. Due to the combination of the shipboard, mooring, glider and Argo measurements a satisfying dataset in the eastern tropical Atlantic could be obtained. But this could only be done due to the extensive observation of the eastern tropical north Atlantic in the recent years (25 research cruises, 1 longtime mooring and several glider deployments).

***Other minor comments***

*In the first paragraph of the introduction, the references to support some general sentences do not seem to me the most appropriate (for example, lines 6, 7 and 8). I do not mean that the argument is fallacious (magister dixit), but I think there are other studies that might have greater authority to support what is mentioned.*

- That is correct. We include other references at page 2 line 6, 7 and 8:

Line 6:

"In particular, the eastern boundary current system close to the Northwest African coast is a region where northeasterly trade winds force coastal upwelling of cold, nutrient rich waters, resulting in high productivity (Bakun et al., 1990; Pauly and Christensen, 1995; Messié et al., 2009; Lachkar and Gruber, 2012) "

Line 7, 8:

"The ETNA region is characterized by a weak large-scale circulation (Mittelstaedt, 1991; Brandt et al., 2015), but pronounced mesoscale variability (here referred to as eddies) acting as a major transport process between coastal waters and the open ocean (Marchesiello et al., 2003; Correa-Ramirez et al., 2007; Capet et al., 2008a; Schütte et al., 2015; Thomsen et al., 2015; Nagai et al. 2015)."

*P4. L 1-6. Time lag for optode sensors is rather long given important differences between glider dives and climbs. How were the optode data from gliders corrected. Page 4 lines 14-15 and 22-23. Aanderaa optodes were really calibrated (I mean to change the calibration constants) using CTD cast or the casts were used to estimate the accuracy of the optodes.*

- We added more information on the time constant problem on page 4 line 23-24:

"All four autonomous gliders were equipped with Aanderaa optodes (3830) installed in the aft section of the devices. A recalibration of the Optode calibration coefficients were determined on dedicated CTD casts following the procedures of Hahn et al. (2014). These procedures also estimates and correct the delays caused by the slow optode response time (more detailed information can be found in Hahn et al. 2014; Thomsen et al., 2015)."

- The CTD casts are used to change the calibration constants of the Aanderaa optodes. We add one sentence at page 4 line 16 to give that information:

"Optode calibration coefficients were determined on dedicated CTD casts and additional calibrated in the laboratory with water featuring 0% air saturation before deployment and after recovery following the procedures described by Hahn et al. (2014): "

*P7. L24 (and 16). Salinity in the core of ACME is mentioned as an important variable, why did you decided not to show it.*

- That is right. We changed figure 5 and substitute the temperature with salinity (see figure 2).

[Figure]

Figure 2: Meridional velocity, salinity and oxygen of an exemplary **a)** CE and **b)** ACME at the CVOO mooring. Both eddies passed the CVOO on a westward trajectory with the eddy center north of the mooring position (CE 20 km, ACME 13 km). The CE passed the CVOO from October to December 2006 and the ACME between January and March 2007. The thick black lines in the velocity plots indicate the position of an upward looking ADCP. Below that depth calculated geostrophic velocity is shown. The white lines represent density surfaces inside the eddies and the thin grey lines isolines of salinity. Thin black lines in the salinity plot mark the vertical position of the measuring devices. On the right time series of oxygen is shown from one sensor available at nominal 120 m depth.

[revised manuscript text omitted]

Florian Schütte 8.9.16 16:59

Florian Schütte 8.9.16 16:59

Florian Schütte 8.9.16 17:00

Florian Schütte 8.9.16 17:07

Florian Schütte 11.8.16 10:26

Florian Schütte 8.9.16 18:31

Florian Schütte 8.9.16 18:32

Florian Schütte 8.9.16 18:33

Florian Schütte 8.9.16 23:38

Florian Schütte 8.9.16 18:30

Florian Schütte 8.9.16 18:50
**[1] nach oben verschoben:** Most eddies are generated near the eastern boundary, Rossby wave dynamics and the basin scale circulation force these eddies to propagate westwards.

Florian Schütte 8.9.16 18:31

Florian Schütte 8.9.16 18:30

Florian Schütte 8.9.16 18:31

**2. Data and methods**

**2.1 In-situ data acquisition**

For our study we employ a quality-controlled database combining shipboard measurements, mooring data and Argo float profiles as well as autonomous glider data taken in the ETNA. For details on the structure and processing of the database see . For this study we extended the database in several ways. The region was expanded to now cover the region from 0° to 22° N and 13° W to 38° W (see Fig. 2). We then included data from five recent ship expeditions (RV Islandia ISL_00314, RV Meteor M105, M107, M116, M119), which sampled extensively within the survey region. Data from the two most recent deployment periods of the CVOO mooring from October 2012 to September 2015 as well as Argo float data for the years 2014 and 2015 were also included. Furthermore, oxygen measurements of all data sources were collected and integrated into the database. As the last modification of the database we included data from four autonomous gliders that were deployed in the region and sampled two ACMEs and one CE. Glider IFM11 (deployment ID: ifm11_depl01) was deployed on March 13, 2010. It covered the edge of an ACME on March 20 and recorded data in the upper 500 m. Glider IFM05 (deployment ID: ifm05_depl08) was deployed on June 13, 2013. It crossed a CE on July 26 and recorded data down to 1000 m depth. IFM12 (deployment ID: ifm12_depl02) was deployed on January 10, 2014 north of the Cape Verde island São Vicente and surveyed temperature, salinity and oxygen to 500 m depth. IFM13 (deployment ID: ifm13_depl01) was deployed on March 18, 2014 surveying temperature, salinity and oxygen to 700 m depth. IFM12 and IFM13 were able to sample three complete sections through an ACME. All glider data were internally recorded as a time series along the flight path, while for the analysis the data was interpolated onto a regular pressure grid of 1 dbar resolution (see also Thomsen et al., 2015). Gliders collect a large number of relatively closely spaced slanted profiles. To reduce the number of dependent measurements, we limited the number of glider profiles to one every 12 hours. All four autonomous gliders were equipped with Aanderaa optodes (3830) installed in the aft section of the devices. A recalibration of the Optode calibration coefficients were determined on dedicated CTD casts following the procedures of (Hahn et al., 2014). These procedures also estimates and correct the delays caused by the slow optode response time (more detailed information can be found in Hahn et al. (2014); Thomsen et al. (2015)). As gliders move through the water column the oxygen measurements are not as stable as those from moored optodes analyzed by Hahn et al. (2014). We thus estimate their measurement error to about 3 $\mu$mol kg$^{-1}$. The processing and quality control procedures for temperature and salinity data from shipboard measurements, mooring data and Argo floats has already been described by Schütte et al. (2016). The processing of the gliders' temperature and salinity measurements is described in Thomsen et al. (2015). Oxygen measurements of the shipboard surveys were collected with Seabird SBE 43 dissolved oxygen sensors attached to Seabird SBE 9plus or SBE 19 conductivity-temperature-depth (CTD) systems. Sampling and calibration followed the procedures detailed in the GO-SHIP manuals (Hood et al., 2010). The resulting measurement error were ≤1.5 $\mu$mol kg$^{-1}$. Within the CVOO moorings, a number of dissolved oxygen sensors (Aanderaa optodes type 3830) were used. Calibration coefficients for moored optodes were determined on dedicated CTD casts and additional calibrated in the laboratory with water featuring 0% air saturation before deployment and after recovery following the procedures described by Hahn et al. (2014). We estimate their measurement error at <3 $\mu$mol kg$^{-1}$. For the few Argo floats equipped with oxygen sensors a full calibration is usually not available and only a visual inspection of the profiles was done before including the data into the database. The different manufacturers of Argo float oxygen sensors specify their measurement error at least better than 8 $\mu$mol kg$^{-1}$ or 5%, whichever is larger. Note

Florian Schütte 5.9.16 15:11

Florian Schütte 5.9.16 15:12

Florian Schütte 5.9.16 15:12

Florian Schütte 8.9.16 22:56

Florian Schütte 8.9.16 22:59

Florian Schütte 26.4.16 13:22

Florian Schütte 12.8.16 11:37

Florian Schütte 26.4.16 13:22

Florian Schütte 7.9.16 14:57

Florian Schütte 7.9.16 14:57

Florian Schütte 7.9.16 14:57

Florian Schütte 7.9.16 14:57

Florian Schütte 7.9.16 15:02

Florian Schütte 7.9.16 15:02

Florian Schütte 7.9.16 15:02

Florian Schütte 26.4.16 13:22

[revised manuscript text omitted]

In the following, an estimate of the contribution of the negative oxygen anomalies of "dead-zone" eddies to the oxygen distribution of the SOMZ is presented. The satellite-based eddy tracking reveals that on average each year 14 (2) CEs (ACMEs) are propagating from the upwelling system near the coast into the SOMZ and dissipate there. By deriving the oxygen anomaly on density surfaces an oxygen loss profile due to "dead-zone" eddies in the SOMZ is derived (Fig. 9). Note that due to the lower oxygen values within the eddies compared to the surrounding waters in the SOMZ, the release of negative oxygen anomalies to the surrounding waters is equivalent to a local (eddy volume) enhancement of the oxygen utilization by -7.4 (-2.4) $\mu$mol kg$^{-1}$ yr$^{-1}$ for CEs (ACMEs) for the depth range of the shallow oxygen minimum in the SOMZ, i.e. 50 to 150 m depth. Instead of describing the effect of the dead-zone eddies on the oxygen consumption an equivalent view is to consider a box model approach for the SOMZ. The basis of this box model is the mixing of high-oxygen waters (the background conditions) with low-oxygen waters (the "dead-zone" eddies). The average oxygen concentrations within the eddies in the considered depth range, i.e. 50 to 150 m depth, are 73 (66) $\mu$mol kg$^{-1}$ for CEs (ACMEs). The average oxygen concentration of the background field averaged over the same depth range (between 50 and 150 m depth) derived from the MIMOC climatology (Schmidtko et al., 2013) is 118 $\mu$mol kg$^{-1}$. This climatological value includes the contribution of low-oxygen eddies. If we now consider the respective oxygen concentrations and volumes of the SOMZ and the eddies (multiplied by their frequency of occurrence per year), we are able to calculate the theoretical background oxygen concentration for the SOMZ without eddies to be 125 $\mu$mol kg$^{-1}$. Naturally due to the dispersion of negative oxygen anomalies, the oxygen concentrations in the SOMZ without eddies must be higher than the observed climatological values. Attributing the difference of these oxygen concentrations on the one hand in the SOMZ without eddies (125 $\mu$mol kg$^{-1}$) and on the other hand the

Florian Schütte 8.9.16 20:24

Florian Schütte 8.9.16 20:25

Florian Schütte 8.9.16 20:25

Florian Schütte 8.9.16 20:25

Florian Schütte 15.8.16 16:18

Florian Schütte 15.8.16 16:18

Florian Schütte 6.9.16 10:54

Florian Schütte 8.9.16 20:26

Florian Schütte 8.9.16 20:26

Florian Schütte 8.9.16 20:26

Florian Schütte 15.8.16 15:53

Florian Schütte 6.9.16 11:16

Florian Schütte 6.9.16 11:17

Florian Schütte 27.4.16 15:21

Florian Schütte 6.9.16 11:17

Florian Schütte 6.9.16 11:17

[revised manuscript text omitted]

Florian Schütte 10.8.16 13:48

Florian Schütte 9.9.16 18:09

---

## Editor Decision (ED1)

August 08, 2016

Dear Dr Schütte and co-authors,

Insightful comments and suggestions made by the two reviewers, together with your willingness to fully consider them, will lead to a much improved revised manuscript. I recommend that the manuscript be published after minor revisions along the lines described in the two author comment supplements.

In addition to the referee's comments and suggestions, I would also like to add a few other suggestions for improvement:

1. P. 8, lines 20-21 of original manuscript: The confidence interval for the CE oxygen trend includes zero (0.10 ± 0.12). Given this, you cannot say that oxygen is decreasing within the CE. I propose that this sentence might be rephrased as "On average the oxygen concentration decreases by about 0.19 ± 0.08 µmol kg$^{-1}$ d$^{-1}$ in the core of an isolated ACME, but has no significant trend in the core of an isolated CE (0.10 ± 0.12 µmol kg$^{-1}$ d$^{-1}$)."
2. Referee # 1's dislike of the term "dead-zone" is shared by many scientists because these low-oxygen waters are certainly not devoid of life. Given this, please make sure that double quotation marks always accompany the expression "dead-zone" in the final version of the manuscript, so that people understand this expression is a just a metaphor.
3. In your response to Referee # 1 (p.8, lines 24-26), you mention your rationale for picking a 40 µmol kg$^{-1}$ hypoxic threshold. Please include this rationale in the revised manuscript.
4. In your response to Referee # 1 (p.18, lines 10-11), I find this proposed new sentence confusing, especially this bit: "analog to the SLA".
5. In Figure 2 of the authors' reply to anonymous referee #1, please add text labels to the black isopycnal contour lines.
6. The new Figure 1 included in your response to Referee # 2 presents useful additional information relative to Figure 6 of your original manuscript. You might like to consider producing a figure that would present SLA, SST, SSS and Chl a composites for cyclones, anticyclones and ACMEs, thus combining the information found in both of these figures.
7. On page 6 of your response to Referee # 2, you wrote that you changed figure 5 of the original manuscript by substituting the temperature with salinity. I suggest that in the revised manuscript, this particular figure should present both temperature and salinity panels in addition to meridional velocity and oxygen. Also, I must say that I preferred seeing the oxygen contours of the original figure 5, as they present more information than only oxygen concentration at a nominal depth of 120 m that you presented in Figure 2 or your response to Referee # 2.

Best regards,

Denis Gilbert